# DCI: Dual-Conditional Inversion for Boosting Diffusion-Based Image Editing

**Zixiang Li[1,2], Haoyu Wang[1,2], Wei Wang[1,2], Chuangchuang Tan[1,2], Yunchao Wei[1,2], Yao Zhao[1,2]***
[1]Institute of Information Science, Beijing Jiaotong University
[2]Visual Intelligence +X International Cooperation Joint Laboratory of MOE

## Abstract

Diffusion models have achieved remarkable success in image generation and editing tasks. Inversion within these models aims to recover the latent noise representation for a real or generated image, enabling reconstruction, editing, and other downstream tasks. However, to date, most inversion approaches suffer from an intrinsic trade-off between reconstruction accuracy and editing flexibility. This limitation arises from the difficulty of maintaining both semantic alignment and structural consistency during the inversion process. In this work, we introduce **Dual-Conditional Inversion (DCI)**, a novel framework that jointly conditions on the source prompt and reference image to guide the inversion process. Specifically, DCI formulates the inversion process as a dual-condition fixed-point optimization problem, minimizing both the latent noise gap and the reconstruction error under the joint guidance. This design anchors the inversion trajectory in both semantic and visual space, leading to more accurate and editable latent representations. Our novel setup brings new understanding to the inversion process. Extensive experiments demonstrate that DCI achieves state-of-the-art performance across multiple editing tasks, significantly improving both reconstruction quality and editing precision. Furthermore, we also demonstrate that our method achieves strong results in reconstruction tasks, implying a degree of robustness and generalizability approaching the ultimate goal of the inversion process. Our codes are available at: https://github.com/Lzxhh/Dual-Conditional-Inversion

## 1 Introduction

Diffusion models have made significant progress in the field of generative artificial intelligence. Among them, latent Diffusion Models (LDMs) [41] perform the diffusion process in a compressed latent space rather than the pixel space, enabling more efficient and high-quality image generation and editing. This architectural design has made LDMs a powerful and flexible backbone for a wide range of downstream tasks, such as text-to-image generation [36, 40, 43], image editing [31, 4, 48, 3], image restoration [29, 51, 54], style transfer [52, 50, 7], *etc*. In the image editing tasks, the editing is achieved by manipulating the diffusion latent representations. However, in most cases, the corresponding latent representation for a given image is not directly available, which means that we must first perform an inversion process to obtain their latent representations.

The earliest inversion method is DDPM [16], and it has inspired the development of numerous related methods [47, 2, 19]. DDPMs add random noise at each timestep, which leads to the loss of information contained in the original image, resulting in poor reconstruction and editing effects. DDIM inversion [45, 10] reformulates the diffusion process to be deterministic as solving an implicit equation under the assumption that consecutive points along the denoising trajectory remain close.

---

*Corresponding author

39th Conference on Neural Information Processing Systems (NeurIPS 2025).

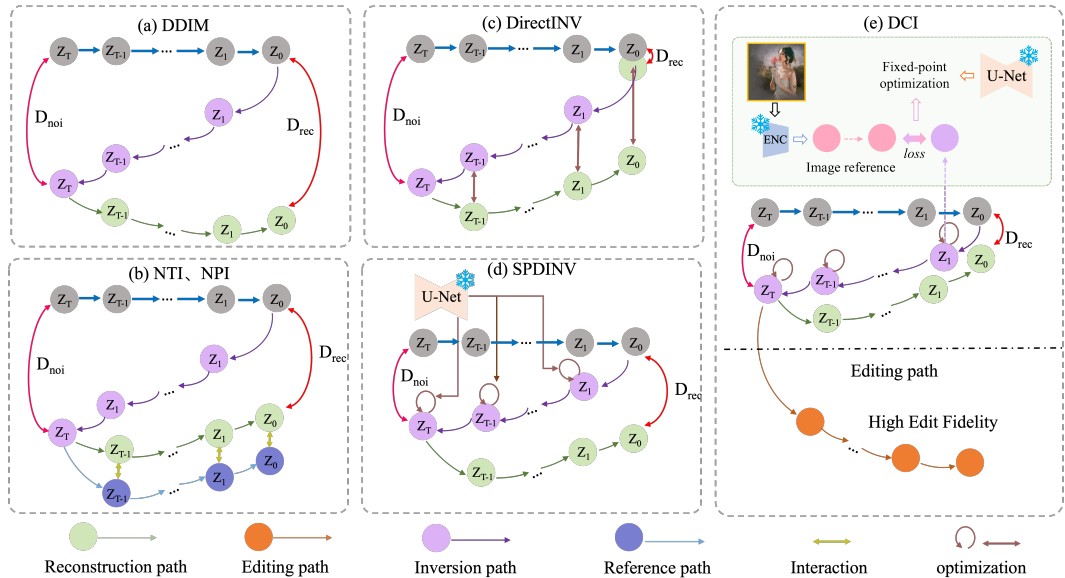

Figure 1: **Pipelines of different inversion methods in diffusion-based image editing.** Each sub-figure illustrates the specific process: (a) DDIM inversion; (b) NTI and NPI; (c) DirectInv; (d) SPDInv; (e) our Dual-Conditional Inversion(DCI). Obviously, DCI significantly reduces both latent noise gap($D_{noi}$) and reconstruction error($D_{rec}$).

However, in practice, especially when using a limited number of denoising steps, this assumption often breaks down, leading to significant inaccuracies in the inversion results. In order to improve the reconstruction effect of DDIM inversion, multiple works have proposed effective optimization methods, such as null-text embedding(NTI) [33] and negative prompt(NPI) [32] in the inversion process. As illustrated in figure 1, both NTI and NPI attempt to reduce the reconstruction gap($D_{rec}$) by optimizing the text embeddings. In the meanwhile, the researchers have proposed some alternative solutions from a non-optimization perspective. For instance, DirectInv [20] introduces a target-aware branch to correct the source branch trajectory, improving reconstruction quality. It performs well especially in terms of content preservation, and it is faster than optimization-based inversion methods. Renoise [13] is based on the linear assumption that the direction from $z_t$ to $z_{t+1}$ can be approximated by the reverse direction from $z_t$ to $z_{t-1}$. By calculating the direction from $z_t$ to $z_{t+1}$ multiple times and taking the average, a more accurate direction from $z_t$ to $z_{t+1}$ could be obtained. SPDInv [25] uses an optimization method to bridge the latent gap on each timestep, but the improvement of reconstruction gap ($D_{rec}$) is limited. Although these methods have achieved certain success, they still face an intrinsic trade-off between reconstruction accuracy and editing flexibility. As illustrated in figure 1, such approaches struggle to reconcile semantic precision with structural consistency, particularly when textual supervision is sparse or ambiguous.

In this work, we present **Dual-Conditional Inversion (DCI)**, a new perspective on diffusion-based image editing that unifies text and image conditioned inversion within a fixed-point optimization framework. DCI addresses this limitation by introducing a dual-conditioning mechanism: it jointly leverages the source prompt $p_s$ and the reference image $x_0$ to guide the inversion process. At the core of our formulation is a two-stage iterative procedure. The first stage, *reference-guided noise correction*, refines the predicted noise at each timestep by anchoring it to a visually grounded reference derived from the source image. The second stage, *fixed-point latent refinement*, imposes self-consistency by optimizing each latent variable $z_t$ as a fixed point of the generative trajectory defined by DDIM dynamics. Formally, we cast inversion as a dual-conditioned fixed-point optimization problem that minimizes two objectives: (1) the discrepancy between the predicted and reference noise vectors across timesteps, and (2) the reconstruction error between the generated image and the original reference. This formulation not only improves inversion stability but also yields latent representations that are inherently editable and semantically aligned.

To sum up, our framework enables a plug-and-play integration with a variety of existing diffusion models, requiring neither retraining nor any modification to the original model. Through extensive experiments across multiple editing tasks, DCI achieves superior reconstruction quality and editing

fidelity when compared to prior inversion baselines. Moreover, we demonstrate that the proposed dual-conditional fixed-point formulation facilitates stable convergence and generalizes well across a wide range of editing scenarios, highlighting the robustness and scalability of the proposed approach.

## 2 Related Work

### 2.1 Image Editing with Diffusion Models

In recent years, a large number of works based on diffusion models in the field of image editing demonstrate significant potential and adaptability across diverse tasks. These methods utilize diverse forms of guidance, such as text prompts, image references and segmentation maps to achieve editing objectives. [24, 22, 8, 17, 27] These advances better enable the ability to maintain editing precision and semantic consistency. The rapid development of diffusion models has significantly improved image generation capabilities. Among them, the widespread use of models such as GLIDE [36], Imagen [43], DALL·E2 [40], and Stable Diffusion(SD) [41] has gradually expanded downstream tasks based on image generation. Prompt-to-Prompt(P2P) [15] modifies cross-attention maps in diffusion models to enable text-driven image editing while preserving spatial structure through localized prompt adjustments. Pix2pix-zero [38] achieves zero-shot image-to-image translation by aligning latent features with text guidance. Plug-and-Play [48] integrates task-specific modules into pretrained diffusion backbones without retraining. MasaCtrl [4] enhances real-time spatial control in diffusion models by injecting mask-guided attention constraints for precise region-specific manipulation. IP-Adapter [56] injects visual features into the attention mechanism, enabling personalized generation without fine-tuning. ControlNet [57] introduces an auxiliary network to condition diffusion models on structural inputs like edges or poses. Some recent efforts have proposed different approaches to improve the precise of image editing from various perspectives [53, 34, 42, 21]. Despite these methods have shown promising results, they often suffer from editing failures due to inversion methods. Our DCI improves upstream inversion to enhance downstream editing fidelity.

### 2.2 Inversion methods of diffusion models

The earliest inversion methods include DDPM [19] and DDIM [45]. DDPM generates high-quality images by progressively adding noise in a forward process and learning the reverse denoising process. [9, 46] Building on this foundation, DDIM introduces a deterministic sampling mechanism. Its near-invertible properties provide a crucial foundation for subsequent image inversion and editing techniques. Researchers have conducted in-depth and extensive studies on the inversion process of diffusion models to achieve both efficiency and precision. Some methods focus on optimizing text embedding [33, 32, 14]. Null-Text Inversion (NTI) [33] adjusts latent encodings and text embeddings to reconstruct the original image. To improve efficiency, Negative-Prompt Inversion (NPI) [32] and its enhancements, including Proximal Guidance [14], have emerged to reduce the reliance on time-consuming optimization processes. EDICT [49], for example, achieves exact invertibility through coupling transformations, while methods like Direct Inversion [20] and Fixed-Point Inversion [30] focus on simplifying the inversion process. The former decouples the diffusion branches, while the latter utilizes fixed-point iteration theory to ensure high reconstruction quality while reducing computational overhead. Many inversion techniques also particularly focus on improving downstream editing tasks [25, 11]. For example, Source Prompt Disentangled Inversion (SPDInv) [25] aims to decouple image content from the original text prompt, enhancing editing flexibility and accuracy. Specialized inversion and editing frameworks have been developed for specific editing needs [26, 44]. Additionally, the concept of inversion has been extended to broader domains [12, 18, 11, 6, 59]. Textual Inversion proposes learning new text embeddings to represent user-specific concepts for personalized image generation [12]. ReVersion [18] further explores learning and inverting relational concepts from images. Meanwhile, works like Aligning Diffusion Inversion Chain [59] focus on generating high-quality image variants by aligning inversion chains.

Although the above methods have solved the reconstruction problem to a certain extent, they may bring artifacts and inconsistent details when applied to editing tasks. Most of the time, they only focus on the text prompt or the original image, but do not integrate them. In our work, we propose a simple but effective method to fuse the text prompt and source image in the form of fixed-point iteration. Our method improves the editing fidelity a lot and shows inspiring results.

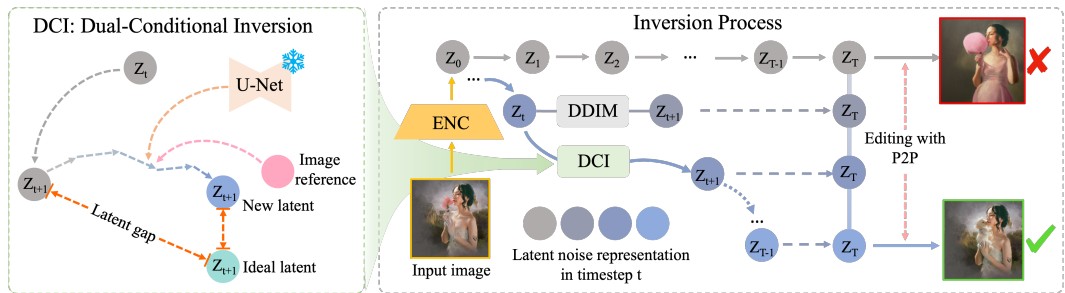

Figure 2: **Inversion process of DCI**. The green box on the left illustrates DCI, which use dual-conditional guidance to reduce the latent gap. The right describes how DCI modifies the inversion process and generate the latent noise code. It also shows our method can improve the editing method.

# 3 Dual-Conditional Inversion

## 3.1 Motivation and Problem Formulation

In most diffusion-based image editing frameworks, the inversion process plays a foundational role: it converts an image to the latent noise representation from which the image can be reconstructed and edited. However, diffusion models inherently lack an explicit and exact inverse process to convert an image back to its corresponding latent noise representation. Ideally, a successful inversion would yield a latent code $z_T$ that faithfully preserves both the semantic content and structural details of the input image, thereby enabling accurate reconstruction and precise downstream editing. However, the information loss caused by repeated noise injection in inversion process makes perfect inversion unattainable, even when auxiliary constraints such as text prompts or reference images are employed.

To analyze the limitations of current inversion strategies, we begin with DDIM (Denoising Diffusion Implicit Models) [45], a deterministic variant of DDPM [16]. DDIM defines a closed-form sampling process that generates a latent image $z_0$ from Gaussian noise $z_T \sim \mathcal{N}(0, \mathbf{I})$ as follows:

$$z_{t-1} = \frac{\sqrt{\alpha_{t-1}}}{\sqrt{\alpha_t}} z_t + \sqrt{\alpha_{t-1}} \left( \sqrt{\frac{1}{\alpha_{t-1}} - 1} - \sqrt{\frac{1}{\alpha_t} - 1} \right) \epsilon_\theta(z_t, t, c), \tag{1}$$

Where $\alpha_t$ denotes the cumulative noise schedule, and $\epsilon_\theta$ represents the noise predicted by a U-Net, conditioned on the current timestep $t$ and a control input $c$ (e.g. , a text prompt). However, using only a text prompt as $c$ is insufficient for accurately reconstructing the original image. Recent methods such as ControlNet [57] and IP-Adapter [56] enrich the conditioning input $c$ with visual features from the original image, thereby improving generation quality. Nevertheless, these methods are often computationally expensive and difficult to integrate into the inversion process. Ideally, inversion requires recovering $z_t$ from a known $z_{t-1}$, which leads to the following "ideal inversion" formula:

$$z_t = C_{t,1} \cdot z_{t-1} + C_{t,2} \cdot \epsilon_\theta(z_t, t, c_{\text{ideal}}), \tag{2}$$

where the coefficients are defined as: $C_{t,1} = \frac{\sqrt{\alpha_t}}{\sqrt{\alpha_{t-1}}}, \quad C_{t,2} = \sqrt{\alpha_t} \left( \sqrt{\frac{1}{\alpha_t} - 1} - \sqrt{\frac{1}{\alpha_{t-1}} - 1} \right).$

However, in practice, this expected inversion is not feasible because the ideal latent $z_t$ is not available when performing the inversion step from $z_{t-1}$. Thus, the DDIM inversion process approximates this update by feeding $(z_{t-1}, t-1, c)$ into the inversion process instead of $(z_t, t, c)$, leading to the practical inversion formula:

$$z_t = C_{t,1} \cdot z_{t-1} + C_{t,2} \cdot \epsilon_\theta(z_{t-1}, t-1, c). \tag{3}$$

This approximation breaks the strict reversibility of the ODE-based formulation and introduces temporal mismatch error between the predicted noise and the actual generative trajectory. Since the diffusion model assumes infinitesimal step size for reversibility (akin to a continuous ODE), using coarse discrete steps and mismatched inputs (i.e., $\epsilon_\theta(z_{t-1}, t-1, c)$ instead of the ideal $\epsilon_\theta(z_t, t, c)$) induces systematic error at each timestep.

If a real image and its corresponding text prompt are given, the image generated directly using the text prompt will be very different from the real image. The reason arises from the inaccuracy of text

prompt and randomness in the generation process. From this perspective, there are also errors in the use of $\epsilon_\theta(z_t, t, c))$ for the inversion process. This error is also accumulated over time, resulting in the final $z_t$ not being well applied to reconstruction and editing. In previous work, SPDInv [25] transforms the inversion process into a search problem that satisfies fixed-point constraints. The pre-trained diffusion model is used to make the inversion process as independent of the source prompt as possible, thereby reducing the gap between $\epsilon_\theta(z_t, t, c))$ and $\epsilon_\theta(z_{t-1}, t-1, c)$. Although SPDInv narrows the gap between $\epsilon_\theta(z_t, t, c))$ and $\epsilon_\theta(z_{t-1}, t-1, c)$. However, in the previous analysis, $\epsilon_\theta(z_t, t, c))$ is not an ideal noise. The ideal noise should not only be separated from the source prompt, but also retain more information of the original image. What needs to be reduced is the difference between $\epsilon_\theta(z_t, t, c_{ideal})$ and $\epsilon_\theta(z_{t-1}, t-1, c)$, and this difference will appear in each inversion process and accumulate in the final output.

To achieve high-fidelity inversion, it is essential to minimize the discrepancy between the predicted noise and the ideal generative direction at each timestep. This requires not only disentangling the inversion process from the source prompt(mentioned in [25]), but also preserving as much information from the original image as possible. Addressing both aspects simultaneously is key to reducing cumulative errors and improving the reconstruction and editability of the inverted latent noise representations in diffusion-based image editing.

## 3.2 Dual-Conditional Inversion (DCI)

To address the limitations of existing inversion methods, we propose Dual-Conditional Inversion (DCI), a novel framework that enhances the latent noise representations in diffusion models. DCI leverages both the original image and text prompt to guide the inversion process, ensuring high-fidelity reconstruction and improved editability. Unlike prior approaches, DCI integrates these into a dual-conditional fixed-point optimization pipeline. The method consists of two key stages: *reference-guided noise correction* that anchors the inversion to the source image, and *fixed-point latent refinement* that ensures self-consistency with the generative process.

### 3.2.1 Reference-Guided Noise Correction

The first stage of DCI introduces a reference-based constraint to align the predicted noise with the source image. At each DDIM timestep $t$, we compute an initial noise estimate conditioned on the source prompt $p_s$:

$$\hat{\epsilon}_{\text{raw}} = \epsilon_\theta(z_t, t, p_s). \tag{4}$$

where $\epsilon_\theta$ is the noise prediction model (e.g., a U-Net) and $z_t$ is the current latent. However, $\hat{\epsilon}_{\text{raw}}$ often deviates from the ideal noise due to the coarse constraint of $p_s$. While this prediction reflects prompt-level semantics, it often deviates from the actual noise corresponding to the input image due to limited grounding provided by textual information alone. To address this, we introduce a visual reference signal by extracting a reference noise vector $\epsilon_{\text{ref}}$ from the source image latent $z_0$, which is obtained via a pretrained VAE encoder $E$. The reference noise is defined as:

$$\epsilon_{\text{ref}} = E(z_0). \tag{5}$$

The $\epsilon_{\text{ref}}$ serves as an anchor to guide the correction of prompt-based noise estimation. To enforce alignment between the prompt-predicted noise and the image-derived reference, we define a reference alignment loss:

$$\mathcal{L}_{\text{ref}} = \|\hat{\epsilon}_{\text{raw}} - \epsilon_{\text{ref}}\|_2 . \tag{6}$$

Equation 6 penalizes the discrepancy between the two noise vectors. A one-step gradient-based correction is then applied to refine the noise prediction:

$$\hat{\epsilon} = \hat{\epsilon}_{\text{raw}} - \lambda \cdot \nabla_{\hat{\epsilon}_{\text{raw}}} \mathcal{L}_{\text{ref}}. \tag{7}$$

where $\lambda$ is a hyperparameter that controls the correction strength. This update adjusts the predicted noise in a direction that reduces its divergence from the reference signal, effectively grounding the inversion in visual structure. As a result, this correction improves reconstruction fidelity and ensures that the denoising trajectory remains semantically and perceptually consistent with the original image, particularly in scenarios where the prompt is ambiguous or underspecified.

### 3.2.2 Fixed-Point Latent Refinement

After correcting the noise estimate, we proceed to update the latent variable $z_t$ using the DDIM inversion formula. This step changes the inversion trajectory from timestep $t-1$ to $t$, based on the corrected noise $\hat{\epsilon}$:

$$z_t = C_{t,1} \cdot z_{t-1} + C_{t,2} \cdot \hat{\epsilon}, \tag{8}$$

where $C_{t,1} = \frac{\sqrt{\alpha_t}}{\sqrt{\alpha_{t-1}}}$ and $C_{t,2} = \sqrt{\alpha_t}\left(\sqrt{\frac{1}{\alpha_t}-1} - \sqrt{\frac{1}{\alpha_{t-1}}-1}\right)$, and $\alpha_t$ is the noise schedule. While this deterministic update follows the DDIM trajectory, it remains sensitive to error accumulation during the inversion process. As such, it may introduce perturbations into the latent dynamics, ultimately affecting reconstruction and editing fidelity. To improve stability and enforce consistency with the forward generative process, DCI introduces a fixed-point refinement step that iteratively corrects the latent by treating it as a fixed-point problem of the DDIM inversion at each timestep. Specifically, we define the latent update function:

$$f_\theta(z_t) = C_{t,1} \cdot z_{t-1} + C_{t,2} \cdot \epsilon_\theta(z_t, t, p_s). \tag{9}$$

The objective is to find a latent $z_t$ such that:

$$z_t = f_\theta(z_t). \tag{10}$$

To achieve this, we minimize the following fixed-point self-consistency loss:

$$\mathcal{L}_{\text{fix}} = \|f_\theta(z_t) - z_t\|_2 \tag{11}$$

We iteratively refine $z_t$ using gradient descent:

$$z_t = z_t - \eta \cdot \nabla_{z_t} \mathcal{L}_{\text{fix}}, \tag{12}$$

where $\eta$ is the learning rate of refinement process. This fixed-point update step is repeated for up to $K$ iterations or until the convergence criterion $\mathcal{L}_{\text{fix}} < \delta$ is satisfied. In practice, our method converges rapidly within a few iterations(usually no more than 10 iterations), which ensures computational efficiency without compromising reconstruction quality. By explicitly enforcing this self-consistency constraint, DCI stabilizes the inversion trajectory and reduces artifacts that arise from misaligned latents. This refinement step not only enhances reconstruction quality but also improves the reliability and flexibility of downstream editing operations.

---

**Algorithm 1** Dual-Conditional Inversion (DCI)

**Input:** Source image latent $z_0$, DDIM steps $T$, source prompt $p_s$, maximal optimization rounds $K$, threshold $\delta$, image guidance strength $\lambda$, fixed-point learning rate $\eta$, reference noise $\epsilon_{\text{ref}}$
**Output:** Inversion noise $z_T$
  1: **for** $t = 1$ to $T$ **do**
  2:     **for** $i = 1$ to $K$ **do**
  3:         Get $z_t$ from $z_{t-1}$ based on (3)
  4:         Predict noise $\hat{\epsilon}_{\text{raw}}$ based on (4)
  5:         Compute $\mathcal{L}_{\text{ref}} = \|\hat{\epsilon}_{\text{raw}} - \epsilon_{\text{ref}}\|_2$
  6:         Apply correction: $\hat{\epsilon} = \hat{\epsilon}_{\text{raw}} - \lambda \cdot \nabla_{\hat{\epsilon}_{\text{raw}}} \mathcal{L}_{\text{ref}}$
  7:         **Update** $z_t$ using $\hat{\epsilon}$
  8:         Calculate $\mathcal{L}_{\text{fix}} = \|f_\theta(z_t) - z_t\|_2$
  9:         Update $z_t = z_t - \eta \cdot \nabla_{z_t} \mathcal{L}_{\text{fix}}$
 10:         **if** $\mathcal{L}_{\text{fix}} < \delta$ *then break* **end if**
 11:     **end for**
 12: **end for**

---

### 3.2.3 Algorithm Summary

The complete Dual-Conditional Inversion (DCI) process is summarized in Algorithm 1. At each DDIM timestep, DCI first performs *Reference-Guided Noise Correction* to obtain a visually grounded noise estimate $\hat{\epsilon}$ by combining prompt-based prediction and reference-derived supervision. Then it is followed by *Fixed-Point Latent Refinement*, which iteratively updates the latent $z_t$ to satisfy a

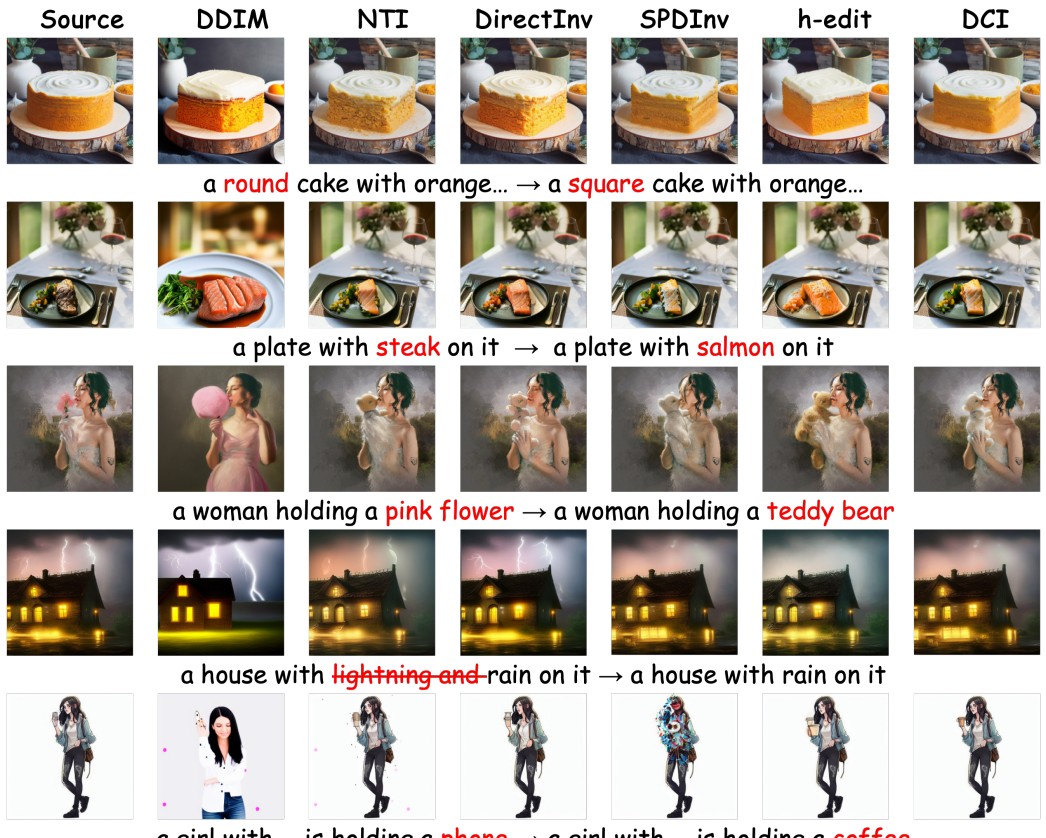

| Source | DDIM | NTI | DirectInv | SPDInv | h-edit | DCI |

a round cake with orange… → a square cake with orange…

a plate with steak on it → a plate with salmon on it

a woman holding a pink flower → a woman holding a teddy bear

a house with lightning and rain on it → a house with rain on it

a girl with … is holding a phone → a girl with … is holding a coffee

Figure 3: **Visual results of different inversion methods with P2P on PIE-Bench.** Each method is identified at the top of its respective column, while detailed editing information appears beneath each corresponding row. DCI(ours) demonstrates significant enhancements over existing methods.

self-consistency condition defined by the DDIM inversion dynamics. The dual conditioning on both the source prompt $p_s$ and the reference image (via $\epsilon_{\text{ref}}$) ensures that the final inverted latent $z_T$ closely approximates the ideal generative noise $z_T^*$, which leads to reliable reconstruction and high-fidelity, better structure-preserving editing.

## 4 Experiments

We conduct extensive experiments to evaluate the effectiveness of Dual-Conditional Inversion (DCI). This section is organized as follows. In Section 4.1, we introduce the datasets, evaluation metrics and experimental settings. Section 4.2 compares DCI with representative inversion methods across multiple aspects quantitatively and qualitatively. In Section 4.3, we investigate how DCI reduces both the latent noise gap and the reconstruction error. Finally, Section 4.4 presents an ablation study to assess the impact of key hyperparameters and design choices.

### 4.1 Experimental Setups

**Evaluation Metrics.** We mainly use DINO score [5], Peak Signal-to-Noise Ratio (PSNR), Mean Squared Error (MSE), Structural Similarity Index (SSIM), and Learned Perceptual Image Patch Similarity (LPIPS) [58] to evaluate the performance of DCI from multiple perspectives. We use the DINO score to evaluate the overall structural similarity of the generated images, while the CLIP score [39] is employed to quantify the alignment between the generated image and the given prompt. For background preservation and image fidelity, we report PSNR, MSE, SSIM, and LPIPS, with all metrics computed specifically over the annotated regions in the dataset from DirecInv [20]. Both the DINO and CLIP scores are calculated over the entire image to capture global consistency, whereas the remaining metrics focus on local quality within specified regions.

Table 1: Performance comparison of inversion-based methods under the Prompt-to-Prompt (P2P) editing engine [11] on PIE-Bench. Metrics include DINO (↓), PSNR (↑), LPIPS (↓), MSE (↓), SSIM (↑), and CLIP (↑). Best and second-best results are highlighted in red and blue, respectively. DCI (ours) achieves the best performance across all metrics.

| Inversion | Editing Engine | DINO↓ $\times 10^3$ | PSNR↑ | LPIPS↓ $\times 10^3$ | MSE↓ $\times 10^4$ | SSIM↑ $\times 10^2$ | CLIP↑ |
|---|---|---|---|---|---|---|---|
| DDIM [45] | P2P | 69.43 | 17.87 | 208.80 | 219.88 | 71.14 | 25.01 |
| NTI [33] | P2P | 13.44 | 27.03 | 60.67 | 35.86 | 84.11 | 24.75 |
| NPI [32] | P2P | 16.17 | 26.21 | 69.01 | 39.73 | 83.40 | 24.61 |
| AIDI [37] | P2P | 12.16 | 27.01 | 56.39 | 36.90 | 84.27 | 24.92 |
| NMG [6] | P2P | 23.50 | 25.83 | 81.58 | 107.95 | 82.31 | 24.05 |
| DirectINV [20] | P2P | 11.65 | 27.22 | 54.55 | 32.86 | 84.76 | 25.02 |
| ProxEdit [14] | P2P | 11.87 | 27.12 | 45.70 | 32.16 | 84.80 | 24.28 |
| SPDInv [25] | P2P | 8.81 | 28.60 | 36.01 | 24.54 | 86.23 | 25.26 |
| *h*-Edit [35] | P2P | 11.17 | 27.87 | 48.50 | 85.40 | 84.80 | 25.30 |
| DCI(ours) | P2P | 6.07 | 29.38 | 33.01 | 21.28 | 87.14 | 25.52 |

**Datasets.** We verifies the effectiveness of our proposed DCI method mainly on the PIE-Bench [20], which comprises 700 images featuring 10 distinct editing types. It provides five annotations on each image: source image prompt, target image prompt, editing instruction, main editing body, and the editing mask. The calculation of region-specific metrics heavily relies on the editing mask, as the editing is expected to occur only within the annotated region. We also use the *COCO2017* [28] to test the application of our method in a wider range of scenarios.

**Other Settings.** In our experiments, we utilize Stable Diffusion v1.4 as the base model with DDIM sampling steps of 50 and a Classifier-Free Guidance (CFG) scale of 7.5. These settings are the same as those used in the baselines. For DCI, we set the hyper-parameters to $K = 5$, $\lambda = 2$, and $\eta = 0.001$. All experiments and validations are conducted on a single NVIDIA RTX 4090 GPU.

## 4.2 Comparisons with Inversion-Based Editing Methods

We compare DCI with several inversion-based methods quantitatively and qualitatively. These methods includes DDIM inversion [45], Null-text inversion (NTI) [33], Negative prompt inversion (NPI) [32], AIDI [37], Noise Map Guidance (NMG) [6], Direct Inversion (DirectINV) [20], Prox-Edit [14], SPDInv [25] and *h*-Edit [35]. We mainly evaluate under the Prompt-to-Prompt (P2P) editing engine on PIE-Bench. As Table 4 shows, DCI (ours) outperforms all methods across DINO, PSNR, LPIPS, MSE, SSIM, and CLIP metrics. Compared to the second-best method, SPDInv, DCI achieves significant improvements, including a 31.1% reduction in DINO (6.07vs.8.81), 8.3% reduction in LPIPS (33.01vs.36.01), and 13.3% reduction in MSE (21.28vs.24.54). At the same time, It is also higher than SPDInv in other metrics(PSNR,SSIM,CLIP). Compared with other methods listed in Table 4, our method has a greater improvement. These results underscore DCI's superior accuracy and robustness for high-fidelity image editing.

Figure 3 presents a visual comparison with the P2P engine. The first row presents cake images frequently used for comparative analysis in existing methods. Most approaches show satisfactory results. In contrast, the second row demonstrates that our method enhances detail representation in salmon. The third row illustrates when modifying features such as hands or mouth, previous methods will fail. However, our DCI achieves this task while maintaining high-quality output. In the fourth row, our method achieves better background color fidelity and reduces lighting artifacts compared to others. The fifth row highlights our method's robust performance in local part editing while preserving overall consistency across other image regions.

**Human Preference Results** For image editing task, human preference is an important part of evaluation metrics. We provide table 2 detailing both user study and human preferences metrics to demonstrate the effectiveness of our approach. For the user study, we collect 40 comparisons from 25 participants (aged 19 to 50). The table shows the mean scores for each participant (min:1, max:5). For human preferences metrics, ImageReward [55] and PickScore [23] are human preference–based

reward models to quantitatively evaluate the quality of image generation and editing. Both of them are higher metrics that represent better performance.

Table 2: Human preference results

|  | DDIM | SPDInv | DCI |
|---|---|---|---|
| Pickscore [23] | 0.4416 | 0.4954 | **0.5547** |
| ImageReward [55] | -0.0120 | 0.1564 | **0.3674** |
| User | 2.14 | 3.48 | **4.10** |

**Time Consumption** The running time of DCI is tied to the number of optimization iterations and the error threshold. However, since our method primarily focuses on nudging the inversion process back onto the correct path, we've empirically found that often only a few optimization steps at specific timesteps are needed to achieve significant improvements. As a result, the additional computational overhead compared with DDIM remains minimal.

Table 3: Comparison of inversion times (in seconds) across different methods.

|  | DDIM | NTI | NPI | AIDI | DirectINV | SPDInv | DCI(ours) |
|---|---|---|---|---|---|---|---|
| Time(s) | 11.55 | 137.54 | 11.75 | 87.21 | 19.94 | 27.04 | 12.13 |

**Results under different editing engines and base models.** Other editing engines and other architectures of diffusion models can also be adopted for our DCI. We use LEDITS++ [2] as the multi-subject editing model and apply it with both DDIM and our DCI method for fair comparison. In our paper, the reported MSE is calculated for non-edited regions, thanks to the availability of appropriate mask annotations within the dataset. However, for multi-subject editing, we could only calculate the MSE between the edited image and the entire original image. Under these conditions, the MSE metric is not always an accurate reflection of editing quality, as DDIM frequently fails to produce any changes or only generates very minor alterations. Beyond multi-subject editing, we also test our method with Stable-Flow, a flow-based diffusion image editing method. The experimental results clearly indicate that our approach significantly enhances performance in flow-based methods as well.

Table 4: Performance under different editing engines and base models.

| Inversion | Editing Engine | DINO↓ $\times 10^3$ | PSNR↑ | LPIPS↓ $\times 10^3$ | MSE↓ $\times 10^4$ | SSIM↑ $\times 10^2$ | CLIP↑ |
|---|---|---|---|---|---|---|---|
| DDIM [45] | LEDITS++ [2] | 21.20 | 21.18 | 136.3 | **76.00** | 83.95 | 19.01 |
| DCI(ours) | LEDITS++ [2] | **12.10** | **21.19** | **127.5** | 125.00 | **84.41** | **21.62** |
| DDIM [45] | Stable-Flow [1] | 19.00 | 24.30 | 91.70 | **37.00** | 91.60 | 23.29 |
| DCI(ours) | Stable-Flow [1] | **4.40** | **24.32** | **68.40** | **37.00** | **92.75** | **23.64** |

Due to the page limit, we provide more visual and quantitative results under different editing engines in the **supplementary material**. We can draw similar conclusions to the above.

### 4.3 Reduction of Noise and Reconstruction Gap by DCI

We conduct experiments and confirm that our method can reduce the gap between noise and reconstruction ($D_{noi}$ and $D_{rec}$ as depicted in Figure 1). We randomly select 100 captions from the PIE-Bench and use Stable Diffusion V1.4 to generate images.

We initialize $z_T$ with a fixed random seed, treating it as the ideal noise input for every image at the initial timestep of the diffusion process. The final generated image serves as a reference for reconstruction accuracy assessment. We visualize and evaluate the performance of our method with DDIM [45] and SPDInv [25].

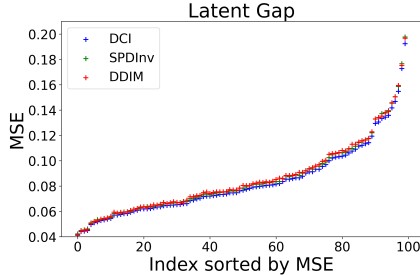

Figure 4: Illustration of Latent Gap.

Table 5: Ablation study on the hyper-parameters of DCI with PIE-Bench.

| Hyper-parameter | $\text{DINO}_{\times 10^3} \downarrow$ | $\text{PSNR} \uparrow$ | $\text{LPIPS}_{\times 10^3} \downarrow$ | $\text{MSE}_{\times 10^4} \downarrow$ | $\text{SSIM}_{\times 10^2} \uparrow$ | $\text{CLIP} \uparrow$ |
|---|---|---|---|---|---|---|
| $K = 2$ | 6.13 | 29.32 | 33.10 | 21.50 | 87.11 | 25.49 |
| $K = 5$ | 6.07 | 29.38 | 33.01 | 21.28 | 87.14 | 25.52 |
| $K = 10$ | 6.17 | 29.29 | 33.17 | 21.56 | 87.12 | 25.51 |
| $\lambda = 1$ | 6.19 | 29.29 | 33.12 | 21.62 | 87.12 | 25.53 |
| $\lambda = 2$ | 6.07 | 29.38 | 33.01 | 21.28 | 87.14 | 25.52 |
| $\lambda = 5$ | 9.29 | 28.25 | 41.05 | 26.18 | 86.26 | 25.38 |
| $\eta = 0.0001$ | 6.72 | 28.80 | 35.93 | 23.72 | 86.70 | 25.50 |
| $\eta = 0.001$ | 6.07 | 29.38 | 33.01 | 21.28 | 87.14 | 25.52 |
| $\eta = 0.01$ | 35.29 | 23.05 | 88.90 | 83.84 | 81.18 | 25.02 |
| **Default** | **6.07** | **29.38** | **33.01** | **21.28** | **87.14** | **25.52** |

For latent gap analysis, we visualize the $z_T$ gap obtained by these methods in figure 4. The concentration of the data shows that our method is closer to the ideal noise. For reconstruction gap evaluation, we use both MSE and CLIP scores. DDIM yields an MSE of $1.32 \times 10^{-4}$, SPDInv achieves $1.21 \times 10^{-4}$, while DCI obtains the lowest error at $1.12 \times 10^{-4}$. The CLIP Scores are 26.91 for DDIM, 26.92 for SPDInv, and 26.94 for DCI. Comparatively, our technique demonstrates superior performance over DDIM and SPDInv based on these metrics.

## 4.4 Ablation Study

Table 5 presents an ablation study on three key hyper-parameters of DCI: the number of optimization rounds ($K \in \{2, 5, 10\}$), the reference-guided noise correction weight ($\lambda \in \{1, 2, 5\}$), and the learning rate ($\eta \in \{0.0001, 0.001, 0.01\}$). The method converges quickly, as even a small number of rounds ($K = 2$) shows competitive results, and performance saturates by $K = 5$. $\lambda = 2$ achieves the best trade-off, while higher values such as $\lambda = 5$ lead to significant degradation across all metrics, indicating over-dependence on inversion constraints. The learning rate $\eta = 0.001$ provides the most stable and effective optimization; both smaller and larger values reduce reconstruction quality, with $\eta = 0.01$ causing severe performance collapse. These results support the choice of the default configuration ($K = 5, \lambda = 2, \eta = 0.001$) as optimal for fidelity and stability.

## 5 Conclusion

In this paper, we introduce Dual-Conditional Inversion (DCI), a novel method that combines both the source prompt and the reference image to guide the inversion process. By formulating inversion as a dual-conditioned fixed-point optimization problem, DCI reduces both latent noise gap and reconstruction errors in diffusion models. Notably, DCI exhibits strong plug-and-play capability: it can be seamlessly integrated into existing diffusion-based editing pipelines without requiring model retraining or architecture modification. Extensive experiments demonstrate that our method achieves superior edit quality on benchmark datasets. Overall, DCI provides a robust, flexible, and easily deployable foundation for future research in diffusion-based tasks.

## 6 Acknowledgements

This research is supported by the National Natural Science Foundation of China (62120106009, 62372033, U24B20179, 92470203, U23A20314), Natural Science Foundation of Beijing, China(No.L252025, No. L242022) and the Fundamental Research Funds for the Central Universities (No. 2024XKRC082).

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

# A  Proof of Dual-Conditional Inversion

## A.1  Stage-Wise Convergence Analysis

The Dual-Conditional Inversion (DCI) algorithm operates in two alternating stages: (1) reference-guided noise correction and (2) fixed-point latent refinement. Since these two stages target separate objective functions and update different variables ($\hat{\epsilon}_{\text{raw}}$ and $z_t$, respectively), their convergence must be analyzed independently.

### A.1.1  Stage 1: Reference-Guided Noise Correction

This stage minimizes the noise alignment loss:

$$\mathcal{L}_{\text{ref}}(\hat{\epsilon}_{\text{raw}}) = \|\hat{\epsilon}_{\text{raw}} - \epsilon_{\text{ref}}\|_2^2, \tag{13}$$

via a one-step gradient descent update:

$$\hat{\epsilon} = \hat{\epsilon}_{\text{raw}} - \lambda \cdot \nabla_{\hat{\epsilon}_{\text{raw}}} \mathcal{L}_{\text{ref}} = (1 - \lambda)\hat{\epsilon}_{\text{raw}} + \lambda \epsilon_{\text{ref}}. \tag{14}$$

This is equivalent to a convex interpolation between $\hat{\epsilon}_{\text{raw}}$ and $\epsilon_{\text{ref}}$, and always satisfies:

$$\mathcal{L}_{\text{ref}}(\hat{\epsilon}) \leq \mathcal{L}_{\text{ref}}(\hat{\epsilon}_{\text{raw}}), \quad \forall \lambda \in (0, 1]. \tag{15}$$

Hence, the correction step is guaranteed to reduce noise misalignment in a single iteration, and can be seen as a contractive projection in noise space.

### A.1.2  Stage 2: Fixed-Point Latent Refinement

Given the corrected noise $\hat{\epsilon}$, we update the latent $z_t$ using the DDIM inversion formula:

$$z_t^{(0)} = C_{t,1} z_{t-1} + C_{t,2} \cdot \hat{\epsilon}. \tag{16}$$

Then, we refine $z_t$ by minimizing the self-consistency fixed-point loss:

$$\mathcal{L}_{\text{fix}}(z_t) = \|f_\theta(z_t) - z_t\|_2^2, \quad \text{where} \quad f_\theta(z_t) := C_{t,1} z_{t-1} + C_{t,2} \cdot \epsilon_\theta(z_t, t, p_s). \tag{17}$$

We apply gradient descent:

$$z_t^{(k+1)} = z_t^{(k)} - \eta \cdot \nabla \mathcal{L}_{\text{fix}}(z_t^{(k)}). \tag{18}$$

Assume:

- $\epsilon_\theta(z)$ is $L$-Lipschitz smooth;
- $f_\theta(z)$ is locally contractive near the solution;
- $\mathcal{L}_{\text{fix}}$ is bounded below.

Then from smooth non-convex optimization theory:

**Theorem A.1** (Local Convergence of Fixed-Point Refinement). *The sequence $\{z_t^{(k)}\}$ satisfies:*

$$\min_{0 \leq k < K} \left\| \nabla \mathcal{L}_{\text{fix}}(z_t^{(k)}) \right\|^2 \leq \frac{2(\mathcal{L}_{\text{fix}}(z_t^{(0)}) - \mathcal{L}_{\min})}{\eta K}, \tag{19}$$

*and converges to a stationary point $z_t^*$ as $K \to \infty$.*

### A.1.3 Alternating Convergence

In DCI, the noise correction step anchors the predicted noise toward a semantically and visually meaningful reference, ensuring that the initialization for the latent refinement step falls within the contraction region of $f_\theta(z)$. Thus, the two-stage alternating optimization benefits from mutual regularization:

- Stage 1 reduces semantic deviation from image-derived noise;
- Stage 2 reduces structural inconsistency via fixed-point updates;

This design avoids the need to jointly optimize a non-separable loss and enables fast convergence with high reconstruction fidelity.

### A.1.4 Additional Analysis

Beyond convergence guarantees, we analyze several critical properties of the two-stage optimization process to better understand the behavior of DCI in practice.

**Stability of the Correction Step.** The reference-guided noise correction step performs a convex interpolation between $\hat{\epsilon}_{\text{raw}}$ and $\epsilon_{\text{ref}}$, ensuring that the corrected noise $\hat{\epsilon}$ remains within the convex hull of the input and the reference:

$$\hat{\epsilon} \in \text{Conv}\left(\hat{\epsilon}_{\text{raw}}, \epsilon_{\text{ref}}\right). \tag{20}$$

This guarantees bounded updates and avoids divergence, even when $\hat{\epsilon}_{\text{raw}}$ contains large errors. Moreover, by interpreting $\lambda$ as a soft trust coefficient, we can view the update as a controllable balance between prompt semantics and visual fidelity.

**Propagation of Residual Noise Error.** Let $\delta_t := \hat{\epsilon} - \epsilon_t^*$ denote the residual noise error at timestep $t$ with respect to the ideal generative noise $\epsilon_t^*$. The DDIM update propagates this noise error linearly into the latent space:

$$z_t = z_t^* + C_{t,2} \cdot \delta_t, \tag{21}$$

where $z_t^*$ denotes the latent corresponding to ideal inversion. Hence, even if $\mathcal{L}_{\text{ref}}$ is not minimized to zero, the resulting latent perturbation is bounded and scales linearly with $\|\delta_t\|$, which DCI attempts to iteratively reduce via fixed-point refinement.

**Editability Preservation.** Unlike optimization methods that overly constrain $z_t$ toward reconstruction, DCI balances reconstruction and generative semantics. The fixed-point loss $\mathcal{L}_{\text{fix}}$ enforces consistency with the model's forward trajectory rather than a hard projection to a reconstruction target, which helps preserve the generative flexibility required for downstream editing. Formally, if $f_\theta(z)$ approximates the forward generative trajectory, minimizing $\|f_\theta(z_t) - z_t\|$ ensures that $z_t$ lies on a semantically meaningful denoising path, rather than collapsing to a static reconstruction point.

**Numerical Robustness.** Empirically, the fixed-point refinement converges within 3–10 iterations under a moderate learning rate $\eta \in [10^{-4}, 10^{-2}]$. As $\nabla\mathcal{L}_{\text{fix}}$ involves only first-order derivatives of the denoiser $\epsilon_\theta$, the update is numerically stable under automatic differentiation and does not amplify high-frequency errors.

**Impact of $\lambda$ and $\eta$ on Optimization Dynamics.** DCI offers explicit knobs to trade off visual grounding ($\lambda$) and convergence aggressiveness ($\eta$). Large $\lambda$ may overfit to the reference signal and degrade semantic consistency; large $\eta$ may induce oscillation or overshoot in latent updates. As shown in the ablation (Table 2), default values $\lambda = 2$ and $\eta = 0.001$ yield a stable equilibrium across editing tasks.

## B  Experimental Results on Different Datasets and Editing Engines

We first present additional results of our method on the PIE-Bench benchmark. As shown in Figure 6, our approach clearly outperforms existing baselines. The red circles highlight undesirable artifacts and imprecise edits introduced by other methods, while our method achieves target edits with high

fidelity. These results demonstrate that our approach excels in terms of editing precision, artifact suppression, and background consistency.

We then evaluate our method on a broader dataset, specifically in *COCO2017* [28], to assess generalization to open-world scenarios. We employ our Dual-Conditional Inversion (DCI) for the inversion stage and adopt P2P [15] as the editing engine. The text prompts are generated by a large language model, with only the desired editing attribute manually modified. As shown in Figure 7, our method maintains strong editing performance even in diverse and unconstrained real-world contexts.

Finally, we examine the compatibility of our inversion method with alternative downstream editing engines. We use the popular Masactrl [4] framework as a representative case. As shown in Figure 8, the results demonstrate that our method performs robustly across different editing pipelines, highlighting its generalizability and adaptability.

## C   Inversion Time Comparison

Despite involving an iterative optimization procedure, our DCI method maintains competitive runtime performance. On a single NVIDIA RTX 4090 GPU, a full DCI inversion-editing cycle takes only average 12.1 seconds per image, which is comparable to the baseline DDIM inversion method [45]. This efficiency is largely due to the lightweight design of our fixed-point refinement and the use of one-step noise correction per iteration. Compared to other methods that involve intensive text

Table 6: Comparison of inversion times (in seconds) across different methods.

| Inversion Method | Inversion Time (s) |
|---|---|
| DDIM | 11.55 |
| NTI | 137.54 |
| NPI | 11.75 |
| AIDI | 87.21 |
| NMG | 16.71 |
| DirectINV | 19.94 |
| ProxEdit | 11.75 |
| SPDInv | 27.04 |
| DCI(ours) | 12.13 |

embedding optimization or complex auxiliary modules, such as Null-text inversion (NTI) [33], Negative prompt inversion (NPI) [32], AIDI [37], Noise Map Guidance (NMG) [6], Direct Inversion (DirectINV) [20], ProxEdit [14], and SPDInv [25]—our method achieves a favorable balance between quality and speed. Some of these baselines require additional optimization rounds or rely on extra network branches. DCI requires only a small number of optimization steps and converges quickly, while still avoiding heavy architectural changes, making it more practical for real-world deployment.

## D   Failure Cases

While DCI significantly improves inversion quality and editing controllability, it still exhibits limitations in certain scenarios. Since our method is designed for an independent optimization method of inversion process without being aligned with downstream editing objectives. Its performance can be adversely affected by the characteristics and limitations of the editing engine itself. In particular, failure cases may arise when the editing model lacks sufficient semantic alignment or spatial precision, resulting in incomplete edits or distorted outputs.

As shown in Figure 5, one failure example occurs when the target edit conflicts with the original content. In this case, the edited image either fails to reflect the desired changes or introduces unwanted artifacts. Such outcomes highlight the dependency of DCI on the quality and specificity of downstream editing models.

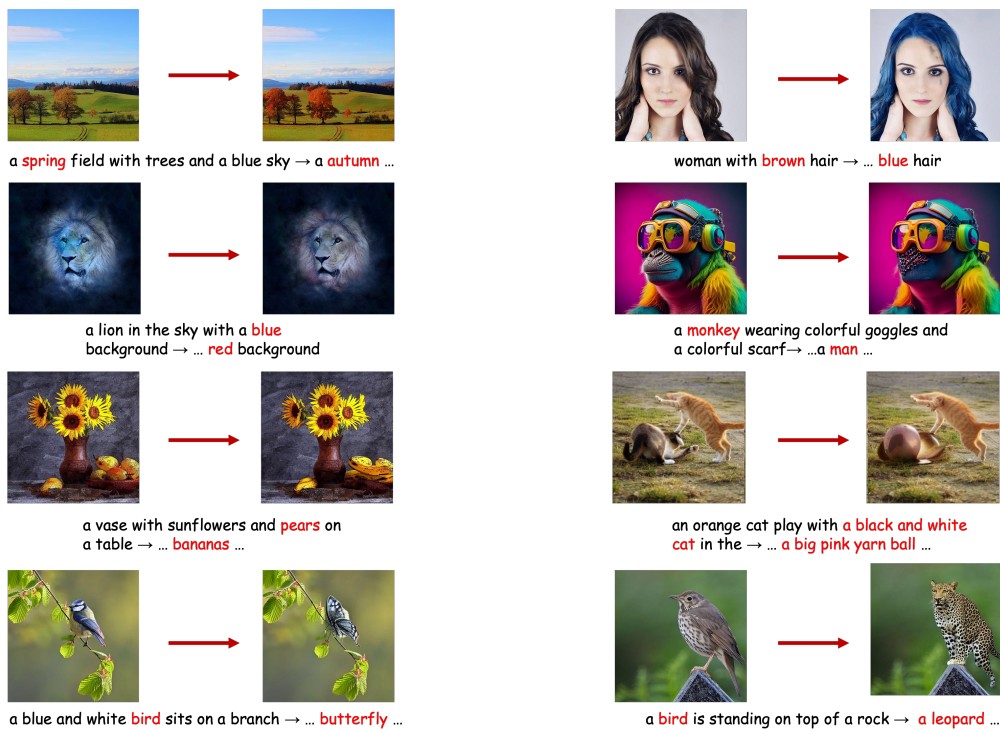

a spring field with trees and a blue sky → a autumn ...

a lion in the sky with a blue
background → ... red background

a vase with sunflowers and pears on
a table → ... bananas ...

a blue and white bird sits on a branch → ... butterfly ...

woman with brown hair → ... blue hair

a monkey wearing colorful goggles and
a colorful scarf→ ...a man ...

an orange cat play with a black and white
cat in the → ... a big pink yarn ball ...

a bird is standing on top of a rock → a leopard ...

Figure 5: Failure cases

# E    Limitations and Future Work

**Limitations.** While our proposed Dual-Conditional Inversion (DCI) framework demonstrates superior performance in diffusion-based image editing, several limitations remain.

First, DCI introduces additional computational overhead due to its dual-stage optimization process, including reference-guided noise correction and fixed-point refinement. Although this leads to increased inference time compared to purely feed-forward inversion methods, we find the overhead to be acceptable in most practical applications, especially those prioritizing editing fidelity over real-time speed. Second, the effectiveness of DCI depends on the quality and semantic alignment of both the source prompt and the reference image. In cases where the prompt is overly ambiguous or poorly aligned with the image content, the dual-conditioning mechanism may lead to suboptimal inversion performance or conflicting guidance signals. Lastly, the extension of DCI to other modalities such as video, 3D scenes, or multi-view data remains unexplored.

**Applicability Conditions**

- **Editing with ambiguous or weak prompts.** The incorporation of reference images enables stable inversion when text prompts alone are insufficient for guiding precise edits.
- **Applications requiring high reconstruction fidelity.** Tasks such as identity preservation, localized image edits, or photo-realistic retouching benefit from the semantic and structural anchoring provided by DCI.
- **DDIM-based architectures.** DCI is currently implemented and evaluated with DDIM. Compatibility with other samplers (e.g., DPM-Solver) may require re-derivation or empirical verification.

**Future Work.** We plan to explore several promising directions: (1) improving computational efficiency through adaptive early stopping or learned refinement modules; (2) extending DCI to higher-resolution pipelines and multi-modal inputs such as text-image-mask triplets; and (3) evaluating the applicability of DCI in diverse generative backbones, including DiT and other transformer-based diffusion models. We also aim to conduct user studies in practical editing tools to assess robustness, usability, and real-world performance.

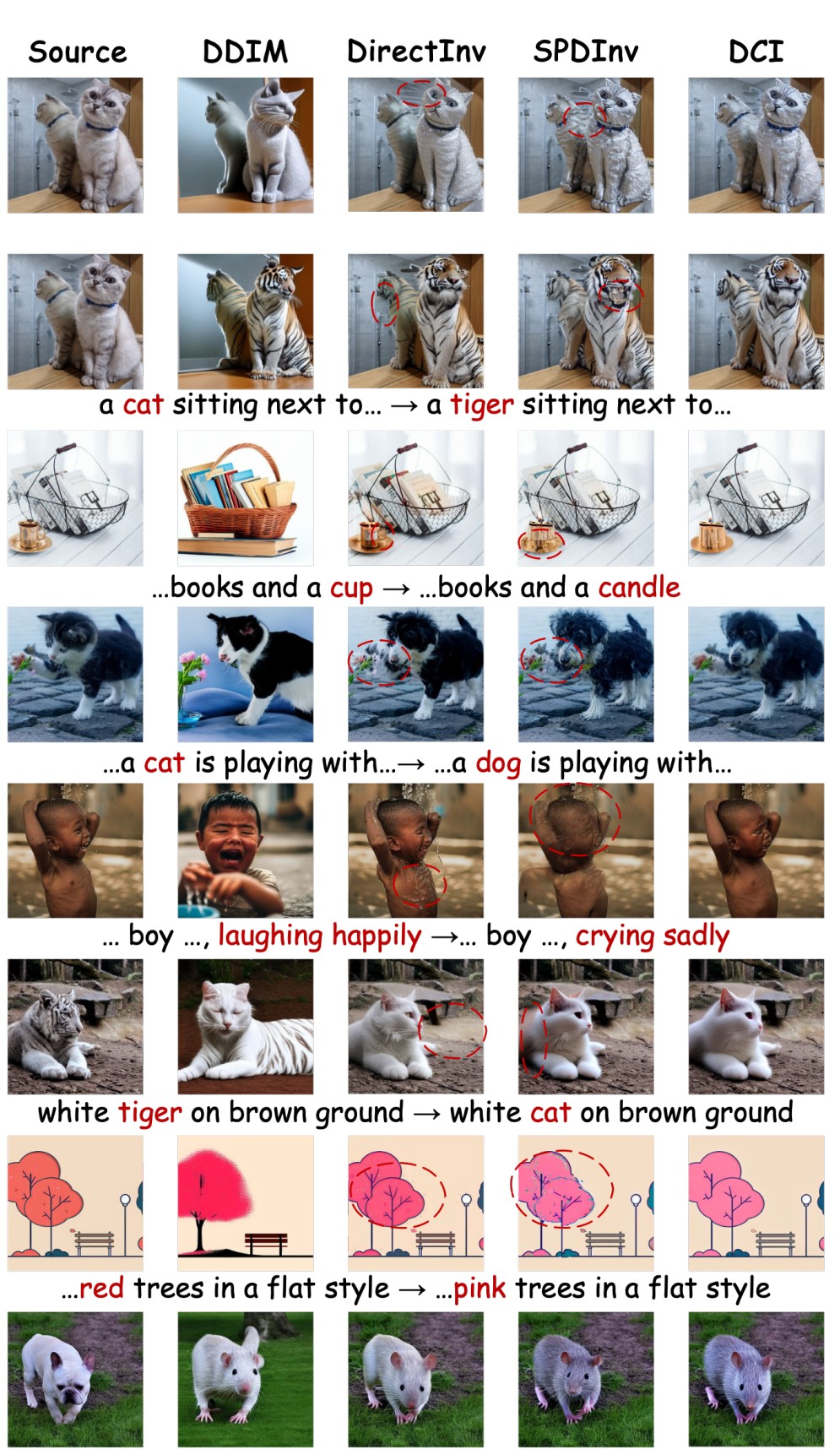

Figure 6: More visual results in PIE-Bench with P2P

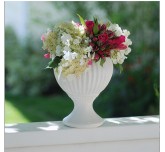 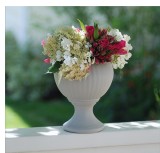

A white vase holds a colorful bouquet of flowers on a sunlit railing → A grey ...

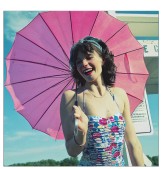 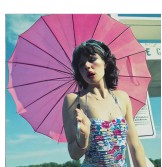

A smiling woman in a swimsuit holds a pink umbrella by the lakeside → A sad ...

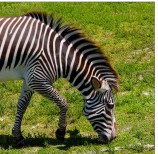 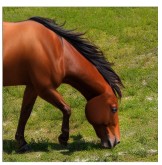

A zebra grazes on green grass under the bright daylight in the open field→ A horse ...

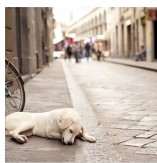 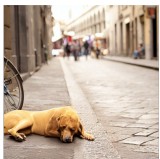

A white dog sleeps peacefully on a quiet street beside a bicycle→ A yellow ...

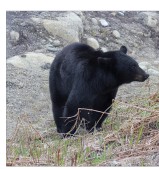 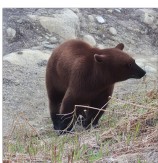

A black bear walks through dry grass and rocky terrain in the wild→ A brown ...

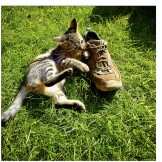 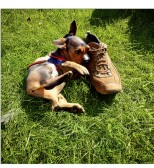

A playful cat bites a brown shoe while lying on green grass→ A playful dog ...

Figure 7: Visual results on *COCO2017* dataset

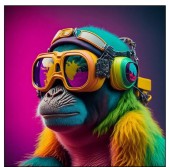 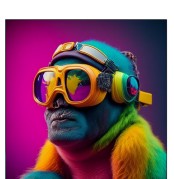

a monkey wearing colorful goggles and a colorful scarf→ ...a man ...

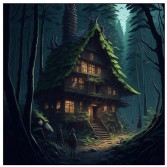 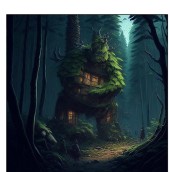

a house in the woods → a monster in the woods

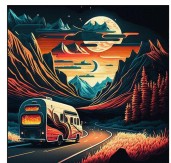 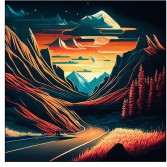

a poster of a bus driving down a road with mountains in the background → a poster of a road ...

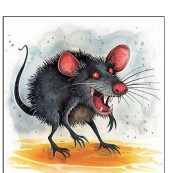

a painting of a rat with red eyes → ...a pig ...

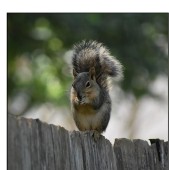 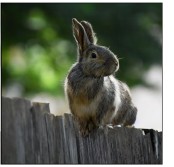

a squirrel is sitting on top of a wooden fence→ a rabbit ...

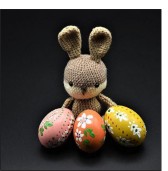 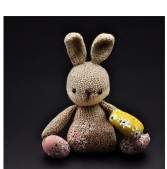

a brownish grey knitted bunny with three painted eggs→ a brownish grey knitted bunny

Figure 8: Visual results of DCI with Masactrl

