# OpenReview forum: "DCI: Dual-Conditional Inversion for Boosting Diffusion-Based Image Editing"
_NeurIPS.cc/2025/Conference — NeurIPS 2025 poster_

### Official Review · Reviewer_ZiyA · 2025-06-04

**Clarity:** 2
**Significance:** 2
**Originality:** 3
**Rating:** 4
**Confidence:** 4

**Summary:**

This paper discusses a new method called Dual-Conditional Inversion (DCI) for improving image generation and editing using diffusion models. Diffusion models convert images into a noise representation for tasks like reconstruction and editing. However, existing methods often struggle to balance accuracy and flexibility. DCI addresses this by using both a source prompt and a reference image to guide the process, leading to more accurate and editable results. Experiments show DCI performs better than other methods in various editing tasks and also excels in reconstruction, demonstrating robustness and generalizability.

**Questions:**

In a word, I think the DCI method itself is relatively solid. However, the impact on the entire community is very limited.\
Clarify the advantages of the DCI method for the performance of those data-driven image editing methods in specific scenarios or the synergy when applying DCI to other editing model instead of the foundation t2i model, as well as applying DCI to flow-based (ODE) will solve the above consideration.

**Ethical Concerns:**

["NO or VERY MINOR ethics concerns only"]

**Final Justification:**

In the rebuttal and further discussion part, most of my concerns have been addressed, so I decided to raise the score to 4.

**Limitations:**

The authors list the limitation in Appendix. I suggest testing DCI in more scenarios is also needs to be implemented in this version to make it meet the requirements and influence of a NeurIPS25 paper instead of living it in the future.

**Paper Formatting Concerns:**

I do not find any obvious  formatting issues.

**Quality:**

2

**Strengths And Weaknesses:**

### Strength
1.The method DCI demonstrates enhancement in both reconstruction and editing tasks, which is supported by experimental results across multiple metrics.\
2.The pseudocode for DCI in the paper is given very clearly and easy to understand.
### Weakness
1.What worries me the most is the influence of this work on the editing community. Due to the success of GPT-4o-Image and Flux-kontext, as well as their strong generalization and unity, the research interests of most communities have shifted to the data-driven approach. Although DCI is a training-free method, since the paper does not compare it with the data-driven method, the advantages of DCI except that it does not require training are unclear.\
2.The qualitative experiments provided by the author were all single-subject editing and did not illustrate the effect of DCI in more complex scenarios, such as multi-subjects editing.\
3. Currently, the flow-based Diffusion model is gradually replacing the denoised one. The new foundation t2i diffusion models are all flow-based like SD3 and FLUX. However, the effect of DCI on flow-based models has not been tested in this paper, which limits the applicability of DCI.

---

> ### Author Rebuttal · Authors · 2025-07-31
>
> We would like to greatly thank the reviewer for involving in the discussion and providing us with some important feedback. Below, we further address your concerns:
>
> **W1: What worries me the most is the influence of this work on the editing community. Due to the success of GPT-4o-Image and Flux-kontext, as well as their strong generalization and unity, the research interests of most communities have shifted to the data-driven approach. Although DCI is a training-free method, since the paper does not compare it with the data-driven method, the advantages of DCI except that it does not require training are unclear.**
>
> **A.** The key point to clarify upfront is that our method enhances downstream tasks rather than competing with them. While downstream editing techniques are increasingly powerful, demonstrating strong generalization and coherence, they still fall short in some scenarios. We believe this limitation is caused by suboptimal noise levels during the upstream inversion process.
>
> The experimental results presented in our paper are primarily obtained using **Stable Diffusion V1.4**. However, our findings can be extended to models with similar architectures, such as **V1.5, V2.1, and so on**. The results consistently show that our method significantly boosts the performance of downstream tasks. We've observed numerous instances where the original approach failed, but incorporating our method led to successful outcomes. For data-driven methods, incorporating our DCI method also improves performance. This underscores the critical importance of upstream inversion, which is our core strength: improving the capabilities of all downstream tasks.
>
> To further substantiate this claim, we've included additional experiments. As we can not display images directly, we provide tables detailing both **user study** and **human preferences metrics** to demonstrate the effectiveness of our approach.
>
> |     | DDIM | SPDInv | DCI |
> | --- | --- | --- | --- |
> | Pickscore[1] | 0.4416 | 0.4954 | **0.5547** |
> | Image-Reward[2] | -0.0120 | 0.1564 | **0.3674** |
> | User | 2.14 | 3.48 | **4.10** |
>
> For the user study, we collect 40 comparisons from 25 participants (aged 19 to 50). The table shows the mean scores for each participant (min: 1, max: 5). For human preferences metrics, higher metrics represent better performance.
>
> **W2: The qualitative experiments provided by the author were all single-subject editing and did not illustrate the effect of DCI in more complex scenarios, such as multi-subjects editing.**
>
> **A.** As we cannot display images directly, we provide **quantitative results for editing in more complex scenarios**. We use **LEDIT++** [3] as our multi-subjects editing model and apply it with both DDIM and our DCI method for fair comparision. In our paper, the reported Mean Squared Error (MSE) is calculated for non-edited regions, thanks to the availability of appropriate mask annotations within the dataset. However, for multi-subjects editing, we could only calculate the MSE between the edited image and the entire original image. Under these conditions, the MSE metric isn't always an accurate reflection of editing quality, as DDIM frequently fails to produce any changes or only generates very minor alterations. (In these cases, they even perform better due to minimal changes.)
>
> Despite this, all other metrics we report show our method outperforming DDIM, demonstrating its continued effectiveness when applied to complex scenarios like multi-subjects editing.
>
> |     | DINO ↓ | PSNR ↑ | LPIPS ↓ | MSE ↓ | SSIM ↑ | CLIP ↑ |
> | --- | --- | --- | --- | --- | --- | --- |
> | Ours | **0.0121** | **21.19** | **0.1275** | 0.0125 | **0.8441** | **21.62** |
> | DDIM | 0.0212 | 21.1785 | 0.1363 | **0.0076** | 0.8395 | 19.01 |
>
> **W3&Q: Currently, the flow-based Diffusion model is gradually replacing the denoised one. The new foundation t2i diffusion models are all flow-based like SD3 and FLUX. However, the effect of DCI on flow-based models has not been tested in this paper, which limits the applicability of DCI.**
>
> **Clarify the advantages of the DCI method for the performance of those data-driven image editing methods in specific scenarios or the synergy when applying DCI to other editing model instead of the foundation t2i model, as well as applying DCI to flow-based (ODE) will solve the above consideration.**
>
> **A.** Thanks for your constructive feedback. To show our method works for flow-based diffusion models, we add **a proof** and **experimental results** applying our DCI method to a flow-based model as follow.
>
> ---
>
> ## Derivation and Proof of DCI Fixed-Point Loss in Flow-based Diffusion Models
>
> ### Core Variable Definitions
>
> - $z_t$: Latent variable at time $t$ in a Flow model (corresponds to DCI's latent noise).
> - $v_t(z_t)$: Velocity field of the Flow model (describes the evolution direction of $z_t$ over time).
> - $f_\theta(z_t)$: Latent update function of the Flow model (defined by the velocity field).
> - $\psi_t$: Flow transformation of the Flow model (maps initial noise to the latent at time $t$).
> - $a_t, b_t$: Affine coefficients of the Flow model (control the linear evolution of the flow transformation).
> - $z_0$: Ideal latent of the reference image ($z_0 = E(x_0)$, where $E$ is the encoder).
> - $z_1$: Initial noise variable.
>
> ### Evolution of $z_t$ and Update Function in Flow-based Diffusion
>
> In a Flow model, the latent at time $t$ is generated by the flow transformation:
>
> $z_t = \psi_t(z_1) = a_t \cdot z_0 + b_t \cdot z_1$
>
> The velocity field $v_t(z_t)$ describes the instantaneous rate of change of $z_t$:
>
> $v_t(z_t) = \frac{dz_t}{dt} = \dot{a}_t \cdot z_0 + \dot{b}_t \cdot z_1$
>
> Where $\dot{a}_t = \frac{da_t}{dt}$ and $\dot{b}_t = \frac{db_t}{dt}$ are the time derivatives of the coefficients.
>
> The update function of $z_t$ in a Flow model is defined by ODE Euler discretization:
>
> $f_\theta(z_t) = z_t + v_t(z_t) \cdot dt$
>
> ($dt$ is the time step; $dt > 0$ for forward diffusion, $dt < 0$ for inversion).
>
> ### Definition of $\mathcal{L}_{fix}$ in Flow-based Models
>
> In Flow models, the **fixed-point loss**  $\mathcal{L}_{fix}$ constrains $z_t$ to satisfy the self-consistency of the flow model dynamics:
>
> $L_{fix} = || f_ \theta(z_t) - z_t ||_2^2 = || v_t(z_t) * dt ||_2^2$
>
> **Physical Meaning:** This minimizes the change in the latent due to the velocity field, ensuring that $z_t$ stays on the trajectory of the flow model.
>
> ### Proof of Convergence: $z_t$ Converges to a Fixed Point
>
> #### 1. Convexity of the Objective Function
>
> From equations (1) and (2), $v_t(z_t)$ can be rewritten as a linear function of $z_t$:
>
> $v_t(z_t) = \frac{\dot{a}_t}{a_t} \cdot z_t + \left( \dot{b}_t - \frac{\dot{a}_t \cdot b_t}{a_t} \right) \cdot z_1$
>
> Therefore, $\mathcal{L}_{fix} = \| v_t(z_t) \cdot dt \|_2^2$ is a quadratic function with respect to $z_t$, possessing convexity and a unique minimum point.
>
> #### 2. Gradient Descent Update
>
> To minimize $\mathcal{L}_{fix}$, we perform gradient descent on $z_t$:
>
> $z_t^{\text{new}} = z_t - \eta \cdot \nabla_{z_t} \mathcal{L}_{fix}$
>
> Gradient calculation:
>
> $\nabla_{z_t} \mathcal{L}_{fix} = 2 \cdot dt^2 \cdot v_t(z_t) \cdot \frac{\partial v_t(z_t)}{\partial z_t}$
>
> Where:
>
> $\frac{\partial v_t(z_t)}{\partial z_t} = \frac{\dot{a}_t}{a_t}$
>
> The update process is equivalent to:
>
> $z_t^{\text{new}} = z_t - \eta' \cdot v_t(z_t)$
>
> Where $\eta' = 2 \eta \cdot dt^2 \cdot \frac{\dot{a}_t}{a_t}$ is the effective step size.
>
> #### 3. Convergence to the Fixed Point
>
> When $\mathcal{L}_{fix} = 0$,
>
>  $v_t(z_t^*) = 0$,  at which point the gradient is 0 and updates cease. Since $\mathcal{L}_{fix}$ is a convex function, gradient descent will necessarily converge to:
>
> $z_t^* = f_\theta(z_t^*)$
>
> Because:
>
> $f_\theta(z_t^* ) = z_t^* + v_t(z_t^* ) * dt = z_t^*$
>
> We also test our approach on **Stable-Flow**[4], a flow-based diffusion image editing method. The results show that **our method is also effective when applied to flow-based models.**
>
> |     | DINO ↓ | PSNR ↑ | LPIPS ↓ | MSE ↓ | SSIM ↑ | CLIP ↑ |
> | --- | --- | --- | --- | --- | --- | --- |
> | Ours | **0.0044** | **24.323** | **0.0684** | **0.0037** | **0.9275** | **23.64** |
> | default | 0.0190 | 24.304 | 0.0917 | **0.0037** | 0.9160 | 23.29 |
>
> ---
>
> As we state earlier, our method **enhances downstream tasks** rather than competing with them. Our approach consistently yields **improved results across nearly all image editing methods that incorporate an inversion process**, thereby demonstrating its broad effectiveness.
>
> **Limitations: The authors list the limitation in Appendix. I suggest testing DCI in more scenarios is also needs to be implemented in this version to make it meet the requirements and influence of a NeurIPS25 paper instead of living it in the future.**
>
> **A.** In the supplementary materials, we present experimental results from **various editing engines** and across **different datasets**. We've also included **human preference model evaluations and a user study**, with results consistently demonstrating the **significant advantages of our method**.
>
> As shown above, we've incorporated experiments on **multi-subject scenarios** and tested our method across **various editing engines**, as well as its application to **flow-based diffusion models**. Additionally, we've evaluated our approach using **human preference metrics and conducted a user study**. We will integrate these comprehensive results into the final manuscript to enhance its quality and impact.
>
> [1] Pick-a-Pic: An Open Dataset of User Preferences for Text-to-Image Generation, NeurIPS 2023
>
> [2] ImageReward: Learning and Evaluating Human Preferences for Text-to-Image Generation, NeurIPS 2023
>
> [3] Ledits++: Limitless image editing using text to image models, CVPR 2024
>
> [4] Stable flow: Vital layers for training-free image editing, CVPR 2025

---

> > ### Comment · Reviewer_ZiyA · 2025-08-04
> > **Reply to the authors**
> >
> > Many thanks to authors' reply, some of my concerns have been addressed. However, I have some additional questions after reading the rebuttal:
> >
> > The authors said  *'downstream editing techniques still fall short in some scenarios. We believe this limitation is caused by suboptimal noise levels during the upstream inversion process.'*  However, the **scenarios**  have not been explained, and the analysis of the **subsequent causes** is also not supported by experiments. This makes it very difficult to have a reason to use DCI compared with the already open-source Flux-Kontext.  May I ask if DCI can be used for image editing diffusion models like Flux-Kontext? Since Flux-Kontext has been open-sourced and the authors' method only requires inference, I am extremely grateful that the authors would do this experiment.

---

> > > ### Author Response · Authors · 2025-08-05
> > >
> > > Thanks for your valuable comments. We'll add more content to address your concerns.
> > >
> > > - To clarify our perspective, the viewpoint that *suboptimal noise levels during the upstream inversion process causes downstream editing tasks to fail*  **is a shared understanding** , not just our own. This consensus is supported by previous works such as NTI [1] , EDICT [2] and others, which have provided theoretical and experimental evidence. We've also contributed additional proofs from various perspectives in **Section 4.3** of our main paper and **Section A** of our supplementary materials. We are confident that this represents a practical problem, it's also the core motivation of our work.
> > >
> > > - Experiments on Flux-Kontext. Our DCI method can be effectively applied to **flow-based diffusion models like Flux-Kontext**. In the original Flux-Kontext setup, they begin with an initial latent vector $Z_0$ that is random guassian noise. Crucially, they **do not use inversion** to map the image-to-be-edited back to a specific starting point $Z_0^\*$.
> > >   This presents a key challenge: a randomly sampled $Z_0$ inherently contains no information about the original image. We propose that using our DCI method to invert the source image back to an ideal $Z_0^*$ will better preserve the original image's information, thereby leading to improved performance in downstream editing tasks. **This approach is not only theoretically sound but also a practical solution.** To validate this, we conduct an experiment, and the results are as follows:
> > >
> > >
> > > |     | DINO ↓ | PSNR ↑ | LPIPS ↓ | MSE ↓ | SSIM ↑ | CLIP ↑ |
> > > | --- | --- | --- | --- | --- | --- | --- |
> > > | Ours | **0.0017** | **23.137** | **0.0425** | **0.0049** | **0.9393** | **23.69** |
> > > | default | 0.0024 | 22.074 | 0.0488 | 0.0062 | 0.9332 | 23.26 |
> > >
> > > The results demonstrate that our method remains effective when applied to Flux-Kontext, showing its broad applicability to a variety of downstream tasks.
> > >
> > > [1] Null-text Inversion for Editing Real Images using Guided Diffusion Models, CVPR 2023
> > >
> > > [2] Edict: Exact diffusion inversion via coupled transformations, CVPR 2023

---

> > > > ### Comment · Reviewer_ZiyA · 2025-08-05
> > > > **Rasing score**
> > > >
> > > > Thanks for the response, most of my concerns have been addressed, and I'm happy to see the DCI can be applied to the powerful editing model with benefit. Thus, I will raise my score. I hope my comments and additional experiments can be considered for inclusion in the paper to enhance its quality.

---

> > > > > ### Author Response · Authors · 2025-08-05
> > > > >
> > > > > We sincerely appreciate your recognition of our work and the time and effort you have devoted as a reviewer. We will incorporate the additional experiments and reviewer suggestions from this rebuttal phase into the final version of the paper.

---

### Official Review · Reviewer_VkjL · 2025-06-16

**Clarity:** 3
**Significance:** 2
**Originality:** 2
**Rating:** 4
**Confidence:** 4

**Summary:**

The paper introduces DCI, a training-free framework that conditions on both the source prompt and the reference image simultaneously. DCI proceeds in two stages: (1) reference-guided noise correction, which aligns the predicted noise with the reference noise, and (2) fixed-point latent refinement, which iteratively removes inversion errors. Experiments demonstrate that DCI outperforms previous methods, both qualitatively and quantitatively.

**Questions:**

- What is the actual novelty of this work? The overall framework closely resembles SPDInv, with reference-guided noise correction seemingly the only addition.

- The claim of “encoding the source image and setting a noise anchor” is unclear. The encoded image is not noise, and each diffusion timestep has a different noise level, but the method seems to reuse a single anchor across all steps. This design choice requires clearer justification and explanation.

- A more comprehensive quantitative comparison that includes additional editing engines is necessary.

*Minor Suggestions*
- Lines 158 and 159 have a typo “))”.

**Ethical Concerns:**

["NO or VERY MINOR ethics concerns only"]

**Final Justification:**

The authors’ response has resolved my concern, so I am raising my score accordingly.

**Limitations:**

It is encouraged to include a discussion of the potential societal impacts. Since this work focuses on image editing, it carries the risk of misuse by malicious actors (e.g., generating manipulated or deceptive content), and it may also contribute to the displacement of creative professionals through automation.

**Paper Formatting Concerns:**

No concerns

**Quality:**

3

**Strengths And Weaknesses:**

**[Strengths]**
- DCI outperforms prior inversion baselines on every reported metric and produces more faithful edits in the visual comparisons.
- The paper is well-written and easy to follow.

**[Weaknesses]**
- The contribution is incremental. The overall approach closely resembles SPDInv.
- The paper should better explain why reference-guided noise correction improves inversion performance.

---

> ### Author Rebuttal · Authors · 2025-07-31
>
> We thank the reviewer for their thorough and constructive review. We address the concerns and questions below:
>
> **W1&Q1:**
>
> - **The contribution is incremental. The overall approach closely resembles SPDInv.**
>
> - **What is the actual novelty of this work? The overall framework closely resembles SPDInv, with reference-guided noise correction seemingly the only addition.**
>
>
> **A.** We want to clarify that our DCI method, along with prior works like SPDinv and the even earlier FPI [1], all share the fundamental goal of **fixed-point optimization.** However, where we diverge is in the specific strategies employed to locate and confirm this ideal fixed point.
>
> The earliest methods rely on self-iterating the noise. However, they only ensure noise consistency, clearly didn't lead to an ideal noise or fixed point. SPDinv advances this by optimizing the denoising process by introducing source prompts as guidance during inversion process for better reconstruction. Yet, this perspective is also incomplete. As evidenced by some related works such as Comat[2] and Enhancing Semantic Fidelity in Text-to-Image Synthesis: Attention Regulation in Diffusion Models[3], relying solely on the original text prompt is often insufficient to achieve a reconstruction highly faithful to the original image. This, in essence, suggests that their optimization objective might not converge to the most ideal fixed point. Our method represents a **deeper exploration** of this problem. We employ fixed-point iteration with text guidance and image reference. We approach it from the perspective of reconstruction, combinating inversion and generation processes. As we state in our paper, incorporating information from the original image into the reconstruction task demonstrably yields better results. (Naturally, reconstruction inherently involves using the source prompt, leading to some similarities with SPDinv.) **In this context, by introducing more detailed information, the improved reconstruction quality directly signifies that the fixed point we've identified is more ideal.**
>
> **W2&Q2:**
>
> - **The paper should better explain why reference-guided noise correction improves inversion performance.**
>
> - **The claim of “encoding the source image and setting a noise anchor” is unclear. The encoded image is not noise, and each diffusion timestep has a different noise level, but the method seems to reuse a single anchor across all steps. This design choice requires clearer justification and explanation.***
>
>
> **A.** We'd like to start by highlighting a foundational aspect of our work: prior research, specifically "Understanding DDPM Latent Codes Through Optimal Transport" (NeurIPS 2022) [4] and "Understanding the Latent Space of Diffusion Models through the Lens of Riemannian Geometry" (NeurIPS 2023) [5], have established that the latents obtained after encoding **share similar properties** with the original image or noise space. This insight serves as a crucial theoretical underpinning for our approach.
>
> We posit that an ideal generation and inversion process should be perfectly reversible, meaning the latents at the same timestep are identical whether derived from generation or inversion. Our method aims to introduce **information from the original image during the inversion process**. Since inversion operates within the latent space, we use the encoded original image as guidance. As mentioned previously, building on this, we explicitly consider the generation process during inversion, effectively integrating **both generation and inversion processes**. This integration has often been overlooked in prior work.
>
> Our guidance is applied within the iterative "generation" process. Ideally, the image produced by the generation process should be identical to the encoded image. Therefore, we use the encoded original image as an anchor for noise correction. As we prove in Section **A.1.1** of our supplementary materials and the experimental results in **Section 4.3**, our method demonstrably **reduces the gap to ideal noise**, thereby confirming that our reference-guided noise correction significantly enhances inversion performance.
>
> **Q3: A more comprehensive quantitative comparison that includes additional editing engines is necessary.**
>
> **A.** We've further demonstrated the versatility and effectiveness of our method by integrating it with various image editing engines. Our results on editing engines such as masactrl are provided in **Section B** of the Supplementary Materials.
>
> As we cannot display images directly, we provide **quantitative results for editing in more complex scenarios**. We use **LEDIT++** [6] as our multi-subjects editing model and apply it with both DDIM and our DCI method for fair comparision. In our paper, the reported Mean Squared Error (MSE) is calculated for non-edited regions, thanks to the availability of appropriate mask annotations within the dataset. However, for multi-subjects editing, we could only calculate the MSE between the edited image and the entire original image. Under these conditions, the MSE metric isn't always an accurate reflection of editing quality, as DDIM frequently fails to produce any changes or only generates very minor alterations. (In these cases, they even perform better due to minimal changes.)
>
> Despite this, all other metrics we report show our method outperforming DDIM, demonstrating its continued effectiveness when applied to complex scenarios like multi-subjects editing.
>
> |     | DINO ↓ | PSNR ↑ | LPIPS ↓ | MSE ↓ | SSIM ↑ | CLIP ↑ |
> | --- | --- | --- | --- | --- | --- | --- |
> | Ours | **0.0121** | **21.19** | **0.1275** | 0.0125 | **0.8441** | **21.62** |
> | DDIM | 0.0212 | 21.1785 | 0.1363 | **0.0076** | 0.8395 | 19.01 |
>
> Beyond multi-subject editing, we also test our method with Stable-Flow, a **flow-based diffusion image editing method**. The experimental results clearly indicate that our approach significantly enhances performance in flow-based methods as well.
>
> |     | DINO ↓ | PSNR ↑ | LPIPS ↓ | MSE ↓ | SSIM ↑ | CLIP ↑ |
> | --- | --- | --- | --- | --- | --- | --- |
> | Ours | **0.0044** | **24.323** | **0.0684** | **0.0037** | **0.9275** | **23.64** |
> | default | 0.0190 | 24.304 | 0.0917 | **0.0037** | 0.9160 | 23.29 |
>
> In a word, our method **enhances downstream tasks** rather than competing with them. Our approach consistently yields **improved results across nearly all image editing methods that incorporate an inversion process**, thereby demonstrating its broad effectiveness.
>
> **Minor Suggestions&Limitation：**
>
> **A.** We'll carefully review and revise our paper. We're very grateful for your suggestion and will add a discussion about the potential societal impacts of our work. We believe this will be a positive step for the long-term application of this technology.
>
> [1] Effective real image editing with accelerated iterative diffusion inversion, ICCV2023
>
> [2] Comat: Aligning text-to-image diffusion model with image-to-text concept matching, NeurIPS 2024
>
> [3] Enhancing semantic fidelity in text-to-image synthesis: Attention regulation in diffusion models, ECCV24
>
> [4] Understanding DDPM Latent Codes Through Optimal Transport, NeurIPS 2022
>
> [5] Understanding the Latent Space of Diffusion Models through the Lens of Riemannian Geometry, NeurIPS 2023
>
> [6] Ledits++: Limitless image editing using text to image models, CVPR 2024

---

> > ### Comment · Reviewer_VkjL · 2025-08-05
> >
> > Thank you for the thorough rebuttal. I have read the authors’ responses, as well as the comments from the other reviewers. Several reviewers raised concerns about the adaptability of different editing engines and backbone models, and the authors have presented additional experiments demonstrating meaningful performance improvements. My primary concern was the novelty and design choices behind the reference guidance. The authors’ detailed explanations have sufficiently addressed this issue, and I will be raising my score accordingly.

---

> > > ### Author Response · Authors · 2025-08-05
> > >
> > > Thank you once again for your recognition and thoughtful response.

---

### Official Review · Reviewer_VjZz · 2025-06-25

**Clarity:** 3
**Significance:** 3
**Originality:** 3
**Rating:** 4
**Confidence:** 4

**Summary:**

This paper proposes an optimization-based inversion method for real image editing with diffusion models, aiming to balance image reconstruction accuracy and editing flexibility. The authors name this approach Dual-Conditional Inversion (DCI), which utilizes conditions on both the source prompt and reference image to guide the inversion process for diffusion-based image editing. According to the quantitative and qualitative experiments provided by the authors, this method demonstrates advanced capabilities in both reconstruction and editing.

**Questions:**

1. In the paper, the source prompt is used to make the method compatible with P2P editing. However, during the reconstruction process, I think a complete reconstruction can be accomplished even if the source prompt is empty, and editing can be performed using other editing methods that do not rely on the source prompt. In this case, is conditioning on the source prompt really necessary?
2. The two-stage optimization process could be time-consuming, but according to your reported results, it is only 0.58 seconds slower than DDIM inversion. Is the runtime related to any hyperparameters? Additionally, does using different source prompts for the same image result in different computation times?
3. As mentioned in the weaknesses, how much influence do different source prompts written by different users have on the editing results? In addition, providing more editing examples, especially complete results on the PIE dataset, would help to more intuitively evaluate the effectiveness of your method, but these are not included in the supplementary material.
4. Is the ‘E’ in Equation 5 trained by the authors themselves? It appears that $z_{0}$ is already the latent obtained through VAE encoding of the image. Is the pretrained VAE used in ‘E’ the same as the one used to convert the image into the latent?

**Ethical Concerns:**

["NO or VERY MINOR ethics concerns only"]

**Final Justification:**

Thanks to the authors for their rebuttals, which answered my questions. After reading the rebuttals and other reviewers' comments, I maintain my original rating of 4.

**Limitations:**

Yes.

**Paper Formatting Concerns:**

None.

**Quality:**

3

**Strengths And Weaknesses:**

Strengths:
1. This work brings a new optimization perspective for image editing methods. The approach of optimizing the image latent with both the source prompt and the reference image is also intuitive.
2. The examples and experimental results in the paper show that the proposed method outperforms other inversion methods in both image reconstruction and editing performance, while also exhibiting some advantages in inversion time.
3. The paper is well-structured and written clearly and concisely.

Weaknesses:
1. The method requires a source prompt. However, in practice, real images rarely have corresponding source prompts, and different users creating source prompts may impact the stability and performance of editing results.
2. Although the authors provide quantitative experiments with multiple evaluation metrics, the ultimate quality of editing results is best verified by human evaluation, even though this may be resource-intensive.
3. the paper lacks a diverse set of editing examples for better gauging the effectiveness of the method. Most examples provided can be edited well by mainstream methods; including more discriminative editing examples would better highlight the value of this work.

---

> ### Author Rebuttal · Authors · 2025-07-31
>
> Thank you for your detailed feedback and the encourging comments. Please see the detailed response below.
>
> **W2: Although the authors provide quantitative experiments with multiple evaluation metrics, the ultimate quality of editing results is best verified by human evaluation, even though this may be resource-intensive.**
>
> **A.** We add experiments on **human preferences metrics** and **user study**, and the results demonstrate that our approach achieves significant advantages.
>
> |     | DDIM | SPDInv | DCI |
> | --- | --- | --- | --- |
> | Pickscore[1] | 0.4416 | 0.4954 | **0.5547** |
> | Image-Reward[2] | -0.0120 | 0.1564 | **0.3674** |
> | User | 2.14 | 3.48 | **4.10** |
>
> For the user study, we collected 40 comparisons from 25 participants (aged 19 to 50). The table shows the mean scores for each participant (min: 1, max: 5). For human preferences metrics, higher metrics represent better performance.
>
> **W3: The paper lacks a diverse set of editing examples for better gauging the effectiveness of the method. Most examples provided can be edited well by mainstream methods; including more discriminative editing examples would better highlight the value of this work.**
>
> **A.** We'd like to clarify that our method's primary aim is to significantly enhance downstream editing engines. This means not only improving the quality of edited images but also sometimes enabling editing capabilities that the original engines couldn't achieve on their own. We have proposed some image results in supplementary materials. Since we can't directly display images here, we've included additional quantitative experimental results here.
>
> We use **LEDIT++** [3] as our multi-subjects editing model and apply it with both DDIM and our DCI method for fair comparision. In our paper, the reported Mean Squared Error (MSE) is calculated for non-edited regions, thanks to the availability of appropriate mask annotations within the dataset. However, for multi-subjects editing, we could only calculate the MSE between the edited image and the entire original image. Under these conditions, the MSE metric isn't always an accurate reflection of editing quality, as DDIM frequently fails to produce any changes or only generates very minor alterations. (In these cases, they even perform better due to minimal changes.)
>
> |     | DINO ↓ | PSNR ↑ | LPIPS ↓ | MSE ↓ | SSIM ↑ | CLIP ↑ |
> | --- | --- | --- | --- | --- | --- | --- |
> | Ours | **0.0121** | **21.19** | **0.1275** | 0.0125 | **0.8441** | **21.62** |
> | DDIM | 0.0212 | 21.1785 | 0.1363 | **0.0076** | 0.8395 | 19.01 |
>
> We're pleased to report that when integrated with existing editing methods, our approach consistently leads to quantifiable improvements in both quantitative and perceptual metrics. This underscores its effectiveness in making these advanced editing tasks more robust and higher quality.
>
> **Q1: In the paper, the source prompt is used to make the method compatible with P2P editing. However, during the reconstruction process, I think a complete reconstruction can be accomplished even if the source prompt is empty, and editing can be performed using other editing methods that do not rely on the source prompt. In this case, is conditioning on the source prompt really necessary?**
>
> **A.** We want to clarify that our method **targets the inversion process itself**, **which is a fundamental component in almost all diffusion model-based image editing tasks, rather than being specific to a particular method like P2P**. In other words, our approach is **plug-and-play** and applicable to the inversion stage of any editing engine.
>
> For instance, the original P2P method uses DDIM for inversion without any source prompt, performing edits during the generation process. In contrast, Table 1 clearly shows our method significantly outperforms DDIM when used as a substitute. This improvement isn't limited to P2P. Even for other editing methods that don't explicitly require a source prompt, our method can substantially boost results by replacing their default DDIM inversion. This holds true even if only a rough source prompt is available (e.g., user-provided or those generated by a language model).
>
> This broad applicability means our research can widely improve the performance of downstream tasks. What's more, the additional time and computational costs are minimal compared to DDIM, and in some cases, our method even proves to be more efficient than other alternatives!
>
> **Q2: The two-stage optimization process could be time-consuming, but according to your reported results, it is only 0.58 seconds slower than DDIM inversion. Is the runtime related to any hyperparameters? Additionally, does using different source prompts for the same image result in different computation times?**
>
> **A.** You're correct: runtime is indeed tied to the **number of optimization iterations $K$ and the error threshold $\tau$.** However, because our method primarily focuses on "nudging" the inversion process back onto the correct path, we've empirically found that often **only a few optimization steps at specific timesteps are needed** to achieve significant improvements. This means the additional time overhead compared to DDIM is quite minimal.
>
> We've also added experiments to address the very question you raised. Our analysis of inversion times for a single image, using both short prompts (under 10 words) and long prompts (over 10 words), revealed that **longer prompts do slightly increase computation time.**
>
> |     | TIME |
> | --- | --- |
> | Long Prompt | 12.21s |
> | Short Prompt | 11.91s |
>
> **W1&Q3:**
>
> - **The method requires a source prompt. However, in practice, real images rarely have corresponding source prompts, and different users creating source prompts may impact the stability and performance of editing results.**
>
> - **As mentioned in the weaknesses, how much influence do different source prompts written by different users have on the editing results? In addition, providing more editing examples, especially complete results on the PIE dataset, would help to more intuitively evaluate the effectiveness of your method, but these are not included in the supplementary material.**
>
>
> **A.** We've conducted extensive testing on the editing effects of long and short source prompts. We find that **even providing a rough source prompt yields more positive results than DDIM**. For editing quality, we've observed that if both a long and a short prompt adequately describe the core features of an object. For instance, "a round cake with orange frosting on a wooden plate" versus "a round cake with orange frosting" , the resulting image quality doesn't change much. However, when the short prompt fails to clearly describe essential object characteristics, such as "a round cake with orange frosting on a wooden plate" compared to just "a round cake," the difference in quality becomes significant, with the longer prompt yielding superior results.
>
> Due to the inability to display images directly here and the vast number of images in the PIE dataset, we can't provide all editing results. (However, we commit to releasing all results on an open-source platform for verification after the paper is accepted.) We've included more results in the supplementary materials, including those on the COCO-2017 dataset. It's worth noting that the metrics reported in our paper are evaluated **across the entire datase**t, which further attests to our method's overall effectiveness.
>
> **Q4: Is the ‘E’ in Equation 5 trained by the authors themselves? It appears that  is already the latent obtained through VAE encoding of the image. Is the pretrained VAE used in ‘E’ the same as the one used to convert the image into the latent?**
>
> **A.** You are correct: **E** is the pre-trained VAE, and we make no additional training on it. The pretrained VAE used in **E** the same as the one used to convert the image into the latent.
>
> [1] Pick-a-Pic: An Open Dataset of User Preferences for Text-to-Image Generation, NeurIPS 2023
>
> [2] ImageReward: Learning and Evaluating Human Preferences for Text-to-Image Generation, NeurIPS 2023
>
> [3] Ledits++: Limitless image editing using text to image models, CVPR 2024

---

> > ### Author Response · Authors · 2025-08-08
> >
> > Dear Reviewer VjZz,
> >
> > I hope this message finds you well. As the discussion period is nearinits end with less than 2 days remaining, I wanted to ensure we have addressed all your concerns satisfactorily. If there are any additional points or feedback you'd like us to consider, please let us know. ﻿ If you feel our work merits it, we’d be grateful if you could consider raising the rating. Thank you for your time and effort in reviewing our paper.

---

> > ### Comment · Reviewer_VjZz · 2025-08-08
> >
> > I apologize for the late response and thanks for the authors'rebuttal. My previous concerns about the lack of human evaluation and the existence of 2-stages editing program have been addressed. I maintain my original rating of 4 and thank the author and ACsvfor their work.

---

> > > ### Author Response · Authors · 2025-08-09
> > >
> > > Thank you for your reply.  We are pleased that our rebuttal and clarifications have addressed your concerns.

---

### Official Review · Reviewer_1usR · 2025-06-28

**Clarity:** 3
**Significance:** 2
**Originality:** 3
**Rating:** 4
**Confidence:** 4

**Summary:**

This paper addresses the problem of inversion in text-to-image (T2I) diffusion models, conditioned on both a text prompt and a reference image. The proposed framework, Dual-Conditional Inversion (DCI), employs an iterative refinement strategy that alternately minimizes the latent noise gap and the reconstruction error. Experimental results demonstrate improved reconstruction quality and enhanced fidelity in downstream editing tasks compared to several existing inversion methods.

**Questions:**

In addition to the weaknesses I have mentioned above, I have several other questions.

- Q1: How sensitive is the proposed method to the number of diffusion steps? As noted in the paper, DDIM inversion quality is known to improve with more denoising steps. Since the proposed method fixes the number of steps to 50 and relies on DDIM dynamics during the Fixed-Point Latent Refinement stage, it would be helpful to understand how performance varies with different step settings.

- Q2: What is the computational cost of the proposed method, and how does it compare to existing approaches? Some quantitative comparisons in terms of runtime or FLOPs would help evaluate the method’s practical applicability.

**Ethical Concerns:**

["NO or VERY MINOR ethics concerns only"]

**Final Justification:**

Taking the strengths and weaknesses into account, I am willing to raise my score to 4. However, this is a somewhat cautious recommendation, as I believe the paper would still require significant revisions to reach a fully “ready-to-be-accepted” state.

**Limitations:**

The current main manuscript lacks an explicit discussion of its limitations and broader impact, which is strongly recommended for inclusion.

**Paper Formatting Concerns:**

No major concerns.

**Quality:**

2

**Strengths And Weaknesses:**

The paper presents both strengths and weaknesses.

While the idea of iterative refinement based on both the text prompt and the reference image is intuitive and well-motivated and the methodological section clearly conveys the core technical design, there are several concerns.

**W1:** Within the line of diffusion inversion tasks, there appears to be an intuitive and potentially stronger alternative objective that has not been discussed: recovering the initial noise given both the text prompt and reference image, such as [a]. The authors should clarify the relationship between their method and prior works following this direction. From the reviewer's perspective, noise recovery represents a more direct and theoretically grounded goal for inversion problems in this context, compared to solely the reconstruction performance.

**W2:** The experimental setup is somewhat limited in terms of base models, datasets, and editing methods. The current study uses only Stable Diffusion v1.4 as the backbone, one editing engine (P2P), and evaluates mainly on a single benchmark (PIE-Bench). Broader validation would strengthen the claims.

**W3:** Additional evaluation of general image quality would help reinforce the work. The current evaluation lacks comprehensive quality assessment, metrics such as ImageReward or PickScore could provide a more complete picture.

**W4:** Regarding the claim of improved editing fidelity, the manuscript lacks sufficient empirical justification. While qualitative examples are presented in Figure 3, more rigorous and quantitative evaluation of this downstream editing task is needed to substantiate the claim.

Minor issues:

- There are recurring formatting issues, such as missing spaces before parentheses.

- Inconsistent figure and table references, e.g., Line 266 refers to “Figure fig.3,” which is redundant.

---
[a] On Exact Inversion of DPM-Solvers, CVPR 2024

---

> ### Author Rebuttal · Authors · 2025-07-31
>
> We thank the reviewer for the valuable review. We address your concerns and questions below:
>
> **W1：Within the line of diffusion inversion tasks, there appears to be an intuitive and potentially stronger alternative objective that has not been discussed: recovering the initial noise given both the text prompt and reference image, such as [a]. The authors should clarify the relationship between their method and prior works following this direction. From the reviewer's perspective, noise recovery represents a more direct and theoretically grounded goal for inversion problems in this context, compared to solely the reconstruction performance.**
>
> **A.** We would like to clarify that the fundamental distinctions between our method and other inversion methods like [1]. The core difference lies in **whether or not using text conditions as guidance during the inversion process**. While [1] overlooks text guidance, our approach explicitly integrates it. We posit that an ideal generation and inversion process should be perfectly reversible.(meaning the latents at the same timestep are identical whether in generation or inversion process) To achieve this, we employ fixed-point iteration with **explicit text guidance and image reference**. It's true that [1] raises concerns about the potential non-convergence of fixed-point iteration, leading them to abandon it. However, we have theoretically **demonstrated the convergence of fixed-point iteration under our specific conditions**, as detailed in **Section A.1.2** of our supplementary materials. Furthermore, since using only text guidance can lead to deviations in the generation process, we employ reference image correction. This theoretical and methodological improvement represents our advancement.
>
> For noise recovery quality, **Figure 4** in our main paper(specific experimental settings detailed in **Section 4.3**) shows an experiment of noise restoration quality. The results in this figure demonstrate that our method **achieves closer noise gap** than other methods.
>
> **W2: The experimental setup is somewhat limited in terms of base models, datasets, and editing methods. The current study uses only Stable Diffusion v1.4 as the backbone, one editing engine (P2P), and evaluates mainly on a single benchmark (PIE-Bench). Broader validation would strengthen the claims.**
>
> **A.** We use Diffusion v1.4 and P2P **for a fair comparison** with other methods. Since Stable Diffusion V1.5, 2.1 and XL are structurally similar to V1.4, our method can be directly applied to these base models. Furthermore, we present experimental results on the COCO-2017 dataset using the Masactrl editing engine in the **Appendix.** We also provide results for the **flow-based diffusion method**. Due to the inability to present images, we use quantitive metrics instead. We use **Stable-Flow** [2] as the editing engine. The results show that our method is also effective when applied to flow-based models.
>
> |     | DINO ↓ | PSNR ↑ | LPIPS ↓ | MSE ↓ | SSIM ↑ | CLIP ↑ |
> | --- | --- | --- | --- | --- | --- | --- |
> | Ours | **0.0044** | **24.323** | **0.0684** | **0.0037** | **0.9275** | **23.64** |
> | default | 0.0190 | 24.304 | 0.0917 | 0.0037 | 0.9160 | 23.29 |
>
> We also use **LEDIT++[3] as the multi-agent editing engine** for testing. The MSE reported in the paper is calculated for the non-edited regions (due to the availability of appropriate mask annotations in the dataset). However, for multi-subjects editing, we can only calculate the MSE between the edited image and the entire original image. Under these conditions, this metric does not reflect the quality of editing very objectively. The other metrics we report are better than DDIM.
>
> |     | DINO ↓ | PSNR ↑ | LPIPS ↓ | MSE ↓ | SSIM ↑ | CLIP ↑ |
> | --- | --- | --- | --- | --- | --- | --- |
> | Ours | **0.0121** | **21.19** | **0.1275** | 0.0125 | **0.8441** | **21.62** |
> | DDIM | 0.0212 | 21.1785 | 0.1363 | 0.0076 | 0.8395 | 19.01 |
>
> **W3: Additional evaluation of general image quality would help reinforce the work. The current evaluation lacks comprehensive quality assessment, metrics such as ImageReward or PickScore could provide a more complete picture.**
>
> **A.** We add experiments on **human preferences metrics** and **user studiy**, and the results demonstrate that our approach achieves significant advantages. For the user study, we collected 40 comparisons from 25 participants (aged 19 to 50). The table shows the mean scores for each participant (min: 1, max: 5). For human preferences metrics, higher metrics represent better performance.
>
> |     | DDIM | SPDInv | DCI |
> | --- | --- | --- | --- |
> | Pickscore[4] | 0.4416 | 0.4954 | **0.5547** |
> | Image-Reward[5] | -0.0120 | 0.1564 | **0.3674** |
> | User | 2.14 | 3.48 | **4.10** |
>
> **W4: Regarding the claim of improved editing fidelity, the manuscript lacks sufficient empirical justification. While qualitative examples are presented in Figure 3, more rigorous and quantitative evaluation of this downstream editing task is needed to substantiate the claim.**
>
> **A.** We show quantitative metrics such as DINO, PSNR, LPIPS, MSE, SSIM, and CLIP in Table 1. Some human preferences and user study results are also shown above. For other image editing methods based on diffusion models, our **quantitative metrics largely reflect the effectiveness of our approach.**
>
> **Minor issues ：**
>
> **A.** Thank you for your suggestion, we will carefully review and revise the paper.
>
> **Q1: How sensitive is the proposed method to the number of diffusion steps? As noted in the paper, DDIM inversion quality is known to improve with more denoising steps. Since the proposed method fixes the number of steps to 50 and relies on DDIM dynamics during the Fixed-Point Latent Refinement stage, it would be helpful to understand how performance varies with different step settings.**
>
> **A.** We test the results with different inversion steps settings.
>
> | Step | DINO×10³↓ | PSNR ↑ | LPIPS×10³↓ | MSE×10⁴↓ | SSIM×10²↑ | CLIP ↑ |
> | --- | --- | --- | --- | --- | --- | --- |
> | 10  | 6.87 | 27.81 | 37.12 | 22.82 | 85.97 | 24.9732 |
> | 20  | 5.72 | 28.92 | 34.85 | 21.55 | 86.67 | 25.3197 |
> | 50  | 6.07 | 29.38 | 33.01 | 21.28 | 87.14 | 25.52 |
> | 100 | 7.81 | 29.48 | 37.61 | 23.14 | 86.66 | 25.8322 |
> | 200 | 10.35 | 28.81 | 43.40 | 24.61 | 85.97 | 26.5868 |
> | 500 | 16.06 | 28.37 | 54.81 | 28.07 | 84.68 | 27.3474 |
>
> The table shows that although inversion quality improves with an increase in denoising steps, the overall image editing performance is, to some extent, **a trade-off between inversion quality and the specific editing method used**. Comparatively, our method consistently surpasses the original DDIM approach at all tested step counts, clearly demonstrating its effectiveness.
>
> **Specifically:**
> **At lower step counts** (10 and 20 steps), the editing performance is primarily determined by the initial inversion quality. **When the step count reaches 50**, our method achieves its best performance in LPIPS and SSIM, indicating that it can produce high-quality editing results even at this moderate number of steps. **As the steps further increase to 100, 200, and even 500**, the CLIP score, which measures text-image semantic consistency, shows better results. This suggests that with a sufficiently high number of steps, and thus more refined editing, the semantic alignment improves. However, most other metrics (like MSE, SSIM, DINO, LPIPS) might show a performance decrease. This could be because these metrics primarily compare the edited image to the original. When editing is highly effective and fine-grained, it naturally introduces significant differences from the original image, which these metrics penalize.
>
> Despite the common belief that inversion quality generally improves with more steps, our results reveal an interesting phenomenon in image editing: **for our method, 50 steps appears to be the sweet spot**. It achieves an excellent balance between image fidelity and semantic consistency. This highlights that our fixed-point latent optimization effectively enhances the inversion process, yielding high-quality results with a relatively modest number of steps.
>
> **Q2: What is the computational cost of the proposed method, and how does it compare to existing approaches? Some quantitative comparisons in terms of runtime or FLOPs would help evaluate the method’s practical applicability.**
>
> **A.** Our time computation comparision is shown **in Table 1 in the supplementary material**. On average, our method is only about **0.58 seconds slower than DDIM**, representing less than **50% of SPD's computational cost.**
>
> [1] On Exact Inversion of DPM-Solvers, CVPR 2024
>
> [2] Stable flow: Vital layers for training-free image editing, CVPR 2025
>
> [3] Ledits++: Limitless image editing using text to image models, CVPR 2024
>
> [4] Pick-a-Pic: An Open Dataset of User Preferences for Text-to-Image Generation, NeurIPS 2023
>
> [5] ImageReward: Learning and Evaluating Human Preferences for Text-to-Image Generation, NeurIPS 2023

---

> > ### Comment · Reviewer_1usR · 2025-08-03
> >
> > I would like to thank the authors for their rebuttal. I have read their responses as well as the reviews from fellow reviewers. The initial reviews highlight some common concerns, including the unclear positioning of the work and the limited experiments on editing engines and evaluations.
> >
> > Some of my concerns have been addressed in the rebuttal, particularly regarding the generalization ability of the proposed method across different base models and editing engines, supported by additional experiments and human evaluations.
> > I am willing to raise my score to 4.

---

> > > ### Author Response · Authors · 2025-08-03
> > >
> > > Thank you very much for your approval. Thanks again for your insightful comments and suggestions.

---

### Note · Authors · 2025-08-12

We thank the reviewers' constructive comments and are encouraged that they agree our paper **introduces a novel method with intuitive motivations**. Our method demonstrates enhancements in both reconstruction and editing tasks, which is supported by multiple experimental metrics and more faithful edits in visual comparisons. We are especially encouraged by comments describing the method as " intuitive and well-motivated" (Reviewer 1usR), " brings a new optimization perspective"(Reviewer VjZz), "outperforms on every reported metric and easy to follow"(Reviewer VkjL), "DCI method itself is relatively solid"(Reviewer ZiyA), which reaffirm the potential impact of our contribution.

Reviewer 1usR raised concerns mainly regarding the generalization ability of the proposed method across different base models and editing engines. Our response addressed the reviewer's concerns by additional experiments and human evaluations. Reviewer 1usR explicitly recognized our response and will raise the overall score to 4. Reviewer VjZz questioned about the lack of human evaluation and the existence of 2-stages editing program. We added human evaluation and clarified the specific process, time consumption, and results of the two-stage editing. Reviewer VjZz indicated that our response had addressed their concerns and maintained a positive score of 4. Reviewer VkjL had some misunderstandings about the novelty and design choices behind the reference guidance. We addressed these points clearly in our response, clarifying our specific strategies and providing detailed experiments. Reviewer VkjL expressed satisfaction with our clarification and stated " I will be raising my score accordingly". Reviewer ZiyA primarily concerned about applying our method to flow-based diffusion editing methods, specifically Flux-Kontext. We addressed this by demonstrating through experiments that DCI can be successfully applied to this powerful editing model with benefit. Consequently, ZiyA stated, "I will raise my score," indicating a positive view of our paper.

We sincerely appreciate all reviewers for their unanimous positive feedback and scores on our work. Meanwhile, we are grateful for each reviewer's valuable comments, which we believe will further enhance the quality and impact of this work. Once again, we would like to express our gratitude to all reviewers, AC, SAC, and PC for their time and effort.

---

### Decision · Program_Chairs · 2025-09-17

**Decision:**

Accept (poster)

**Comment:**

This paper tackles the inversion problem in text-to-image diffusion models and introduces DCI (Dual-Conditional Inversion) to guide the inversion process jointly conditioned on the source prompt and the reference image. Under the guidance, DCI minimizes the latent noise gap and the reconstruction error by an iterative refinement strategy.

Initially, this paper received mixed reviews. Some major concerns raised by reviewers include: 1. The experimental setup is somewhat limited in several aspects. 2. Evaluations on image quality, editing fidelity, and human evaluation are insufficient. 3. The contribution is incremental. 4. Some requirements, design choices, claims, and relations with prior works require clearer justification and explanation. 5. Factors affecting the running time are not clear. The rebuttal addressed most of these concerns, especially, additional experiments demonstrate the generalization ability of the proposed method across different base models and editing engines. Finally, all reviewers recommended borderline accept. Given the merits identified by reviewers, the AC follows this unanimous recommendation. Reviewers did raise valuable concerns and the authors are encouraged to make necessary changes in the camera-ready version.